# MORE THOUGHT, LESS ACCURACY? ON THE DUAL NATURE OF REASONING IN VISION-LANGUAGE MODELS

**Xinyu Tian**[†§], **Shu Zou**[†§], **Zhaoyuan Yang**[*], **Mengqi He**[†], **Fabian Waschkowski**[‡§],
**Lukas Wesemann**[‡§], **Peter Tu**[*], **Jing Zhang**[†],
[†]Australian National University    [‡]University of Melbourne    [*]GE Research    [§]Maincode
`firstname.lastname@anu.edu.au, firstname.lastname@ge.com`

## ABSTRACT

Reasoning has emerged as a pivotal capability in Large Language Models (LLMs). Through Reinforcement Learning (RL), typically Group Relative Policy Optimization (GRPO), these models are able to solve complex tasks such as mathematics and code generation. Building on these advances, recent research has sought to extend reasoning to Vision-Language Models (VLMs), yielding promising results across diverse visual tasks. Despite this progress, our study uncovers the dual nature of multimodal reasoning: while it substantially enhances logical inference and facilitates performance on challenging problems, longer reasoning length may gradually impair perceptual grounding, leading to recognition failures on otherwise basic visual questions. Through further analysis, we attribute this phenomenon to visual forgetting, wherein prolonged reasoning length causes models to disregard visual input. To address this, we propose VISION-ANCHORED POLICY OPTIMIZATION (VAPO), a simple yet effective method that explicitly steers the reasoning process toward visually grounded trajectories. Our result model, `VAPO-Thinker-7B`, significantly strengthens the model's reliance on visual information and achieves new state-of-the-art results on various benchmarks.

## 1   INTRODUCTION

Reasoning has long been recognized as an essential capability of Large Language Models (LLMs). Early approaches, such as chain-of-thought (Wei et al., 2022; Kojima et al., 2022), encourage models to produce step-by-step explanations before arriving at an answer, whereas more recent advances have introduced aha moments (Guo et al., 2025; Muennighoff et al., 2025), characterized by cognitive behaviors such as self-reflection and verification. These developments have enabled models to address increasingly complex problems, *e.g.*, mathematics and coding. Consequently, the community has recently sought to integrate reasoning into Vision-Language Models (VLMs). By leveraging Reinforcement Learning (RL) such as Group Relative Policy Optimization (GRPO) (Shao et al., 2024), researchers have trained models to think while tackling challenging multimodal tasks, *e.g.*, geometry or navigation (Yang et al., 2025c; Huang et al., 2025). Remarkably, this integration has also demonstrated promising results on traditional visual tasks, *e.g.*, classification and detection (Liu et al., 2025; Shen et al., 2025), while yielding stronger generalization (Chu et al., 2025).

Building on these successful practices, reasoning has emerged as a powerful and seemingly universal strategy for addressing a wide range of visual tasks. However, its limitations remain largely unexplored. Early doubts are raised by Li et al. (2025), which find that in certain scenarios, models trained with explicit reasoning result in only marginal gains compared to direct answering. More recently, Xia et al. (2025) observe a counter-intuitive trend: unlike in text domains, RL in multimodal settings often exhibits a collapse. As accuracy improves during training, the reasoning process gradually shortens, suggesting that models tend to favor brief, less structured responses. These findings lead us to ask: *Is reasoning truly a consistent performance booster for vision-language models?*

Motivated by this question, we perform a zoom-in analysis of how the reasoning process impacts accuracy of existing VLMs. Beyond the standard settings of direct answering and full-step reasoning, we introduce an early decision mode, where the model is prompted to halt reasoning at selected logical breakpoints and produce an immediate answer. This setup allows us to measure the contribution

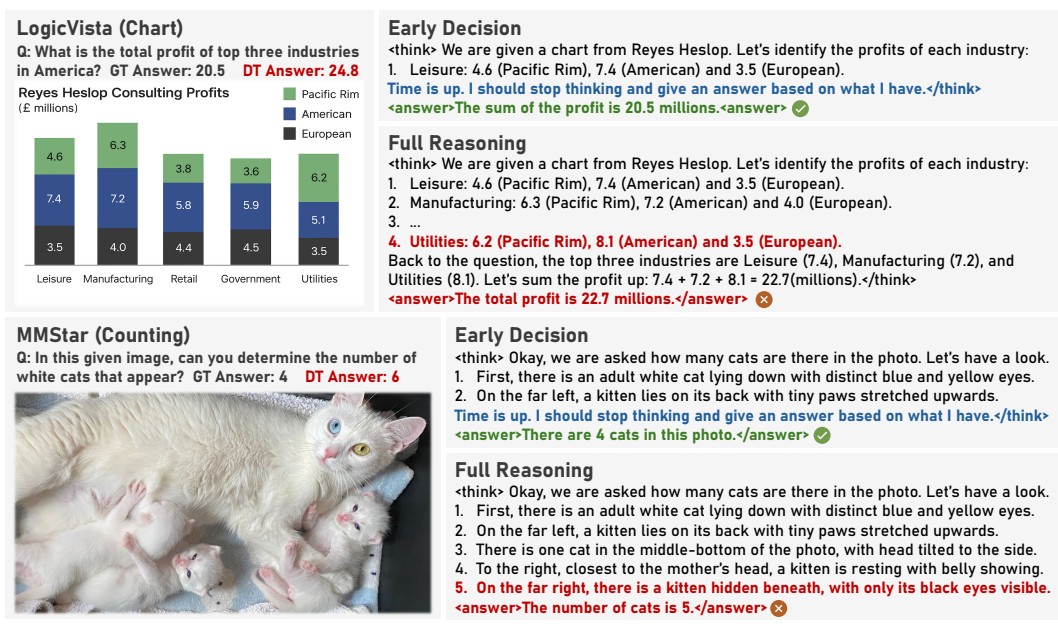

Figure 1: The examples where more reasoning yields less accuracy. We select Vision-R1 (Huang et al., 2025) as a representitive multimodal reasoning model and evaluate on established benchmarks. Given the ground truth (GT) answer, we examine three settings, *i.e.*, direct (DT) answering, full reasoning, and early decision. Correct and incorrect responses are highlighted in green and red, respectively, while designed prompts for early decision are indicated in blue.

of each reasoning segment to the final outcome. Interestingly, our finding reveals that while early-stage reasoning may offer substantial gains, its advantages tend to saturate and may even reverse in later steps. To understand this phenomenon, we conduct an error analysis of the reasoning content in failure cases and find that perception errors, where the model incorrectly interprets visual details in the image, constitute the dominant category, accounting for over 50%. Moreover, the majority of these errors can be rectified with a shorter reasoning via early decision, indicating the model is initially capable of producing correct answers but is ultimately misled by extended reasoning. For instance, as shown in Fig. 1, the early stage of reasoning paths, *i.e.*, early decision, enables the model to arrive at correct answers, whereas prolonged thoughts, *i.e.*, full reasoning, often lead to errors such as chart misreading or hallucination. This indicates the dual nature of reasoning: while it strengthens logical inference and facilitates complex problem-solving, longer reasoning length may weaken perceptual grounding, leading to susceptibility on otherwise straightforward visual questions.

This decline in perceptual capability constitutes a major bottleneck for multimodal reasoning. To further investigate its underlying causes, we examine how attention to visual tokens evolves throughout the reasoning process. Our analysis reveals a marked decrease in visual attention as reasoning progresses, suggesting the emergence of reasoning may inadvertently reduce the model's reliance on visual input, a phenomenon we refer to as visual forgetting. To validate this hypothesis, we propose two interventions: 1) visual replay, which reintroduces the input image to models at regular intervals during reasoning; 2) focus prompt, which prompts models to attend to the visual input at selected steps. Both methods are proven to alleviate performance degradation induced by reasoning, providing evidence that visual forgetting is a central factor constraining reasoning potentials.

While the two aforementioned remedies can partially alleviate visual forgetting, they incur substantial computational overhead during inference and fail to rectify the underlying behavioral deficiencies of existing reasoning models. To address this, we propose VISION-ANCHORED POLICY OPTIMIZATION (VAPO), a simple yet effective policy gradient algorithm that explicitly guides the reasoning process toward a visually grounded trajectory. The key idea is to embed a sequence of visual anchors throughout the reasoning path. At each anchor, the model's perceptual capability is assessed by evaluating its responses to a set of primitive visual claims. Beyond standard outcome-based rewards such as accuracy and format, we introduce perception reward, which quantifies the model's overall perceptual grounding during reasoning by aggregating scores across all anchor points. Experimental results demonstrate that our result model, VAPO-Thinker-7B, significantly enhances

the model's reliance on visual input during reasoning and achieves average gains of $2 \sim 4\%$ over strong baselines on established benchmarks, setting a new state of the art.

In summary, our contributions are as follows:

- Despite the promise of multimodal reasoning, we identify a pronounced dual nature, where reasoning enhances logical inference capability at the expense of perceptual accuracy as the length grows, constituting a major bottleneck to the overall effectiveness.
- We further examine this side effect of reasoning and empirically demonstrate that it stems from visual forgetting, where prolonged reasoning length gradually reduces the model's reliance on visual input, leading to a substantial increase in perceptual failures.
- We propose `VAPO`, a straightforward yet effective policy gradient algorithm designed to strengthen the model's dependence on visual information during reasoning, and demonstrate its effectiveness through experimental validation.

## 2 RELATED WORK

**Reasoning in Large Language Models**. Conventional research has posited a trade-off between interpretability and accuracy (Wang et al., 2020; Koh et al., 2020; Yao et al., 2023b). The advent of reasoning, however, has emerged as a notable exception to the paradigm. Early works demonstrate that prompting LLMs (Wei et al., 2022; Yao et al., 2023a; Zou et al., 2025a; Tian et al., 2024) or applying Supervised Fine-Tuning (SFT) (Yang et al., 2025a; Cai et al., 2024) to encourage step-by-step explanations prior to answering could lead to substantial gains. These findings underscore the critical role of constructing large-scale, high-quality chain-of-thought trajectories. More recently, the emergence of RL-based methods such as GRPO (Shao et al., 2024) has diminished the reliance on manually crafted reasoning paths. With only a limited number of examples, models are now capable of autonomously discovering optimal reasoning strategies and even exhibiting advanced behaviors such as self-reflection and verification (Yu et al., 2025; Zheng et al., 2025). Consequently, Test-Time Scaling (TTS) (Muennighoff et al., 2025; Snell et al., 2025; Qu et al., 2025; Yao et al., 2025) has become a standard practice in LLMs, driven by the belief that longer or more elaborate reasoning consistently yields better performance. However, our empirical analysis challenges the universality of this assumption, revealing that TTS does not necessarily generalize to multimodal settings, particularly regarding its implications for VLMs.

**Reasoning in Vision-Language Models**. Building on the success of reasoning in LLMs, a growing line of research has sought to extend this capability to VLMs (Wang et al., 2025b; Meng et al., 2025; Wang et al., 2025a; He et al., 2025; Tian et al., 2025b;a; Zou et al., 2025b). Two predominant approaches have emerged: one leverages limited high-quality trajectories for cold-start via SFT, followed by RL (Peng et al., 2025; Yang et al., 2025b; Tan et al., 2025; Feng et al., 2025), while the other bypasses SFT entirely, employing RL alone to discover optimal reasoning paths (Liu et al., 2025; Wang et al., 2025d). Both approaches have yielded promising results on various visual tasks, particularly those requiring complex logical inference. However, recent findings indicate that, in certain scenarios, models trained with explicit reasoning offer little or no gains over direct answering (Li et al., 2025). Moreover, training dynamics reveal that as accuracy increases, the length of generated reasoning tends to diminish, implying a preference for shorter and more concise responses (Xia et al., 2025). These preliminary findings indicate that reasoning may not be a free lunch in the development of VLMs. To the best of our knowledge, our work presents the first systematic investigation into the double-edged nature of reasoning in VLMs, identifying visual forgetting as a key limitation and proposing effective solutions to mitigate this issue.

**Forgetting in Vision-Language Models**. In fact, visual forgetting has long been a well-known yet insufficiently addressed issue. Since the emergence of VLMs, researchers have identified a pronounced text bias in these LLM-backboned models, where responses are predominantly guided by textual input while visual cues are largely ignored (Chen et al., 2024a; Fu et al., 2024). At the time, this modality imbalance is less apparent, as early VLMs typically generate short outputs and lack sophisticated reasoning capabilities. Consequently, most prior efforts focus on test-time remedies such as contrastive decoding (Leng et al., 2024; Wang et al., 2024c) or attention reallocation (Tu et al., 2025; Gong et al., 2024) to circumvent visual forgetting. However, with the recent rapid progress in multimodal reasoning, VLMs have increasingly shifted from generating short answers to produc-

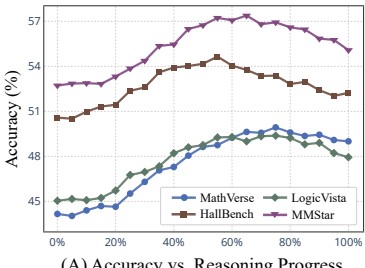 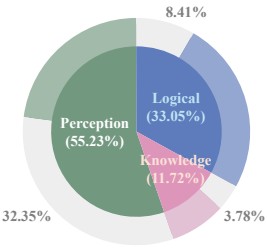 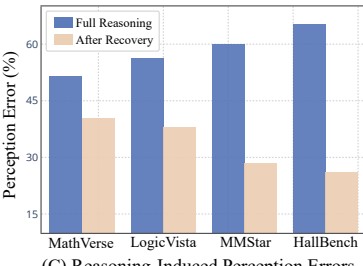

(A) Accuracy vs. Reasoning Progress     (B) Error Category Distribution     (C) Reasoning-Induced Perception Errors

Figure 2: The dual nature analysis of reasoning. In (A), we depict how accuracy evolves throughout the reasoning process. In (B), we show the distribution of error categories under full reasoning (inner ring), alongside the proportion recoverable via early decision (outer ring). In (C), we further present the initial ratio of perception errors on different benchmarks (left bar), as well as the remaining errors that persist after recovery (right bar). It is important to note that recoverable ratio is a loose metric based on a series of early decisions along the reasoning trajectory rather than a single evaluation.

ing extended, step-by-step explanations. This shift has inadvertently amplified the forgetting issue, limiting the benefits of reasoning in multimodal contexts. To address this, we propose `VAPO`, a training-based method that explicitly reinforces the model's perceptual grounding, effectively alleviating visual forgetting, and achieving a new state of the art across established benchmarks.

## 3 THE DUAL NATURE OF REASONING

### 3.1 NO-FREE-LUNCH DILEMMA: LOGIC VS. PERCEPTION

We first explore how accuracy evolves throughout the reasoning process in existing multimodal reasoning models. Specifically, we select three representitve VLMs, *i.e.*, R1-OneVision (Yang et al., 2025c), VLAA-Thinker (Chen et al., 2025) and Vision-R1 (Huang et al., 2025), spanning both SFT-then-RL and RL-only training pipelines. We evaluate models on reasoning-centric benchmarks encompassing MathVerse (Zhang et al., 2024) and LogicVista (Xiao et al., 2024), and vision-intensive benchmarks including MMStar (Chen et al., 2024b) and HallusionBench (Guan et al., 2024). To enable fine-grained analysis, we introduce an early decision mode, in which the model is prompted to terminate reasoning at intermediate logical boundaries such as commas, periods or line breaks, and provide an immediate answer, as illustrated in Fig. 1. This setting allows us to quantify the contribution of each reasoning segment to the overall task performance.

**Longer reasoning does not guarantee better performance**. As shown in Fig. 2 (A), we leverage early decision to control the reasoning length and monitor accuracy trend while reasoning progresses from $0\%$, *i.e.*, direct answering to $100\%$, *i.e.*, full reasoning. In the early stages, reasoning yields substantial gains across benchmarks, consistent with prior works. However, as reasoning continues, these gains tend to plateau and even begin to reverse, leading to a decline in accuracy. This dual effect is particularly pronounced on benchmarks such as MMStar and HallusionBench, where accuracy drops by over $2\%$ from the peak, almost offsetting the original benefits of reasoning.

The above analysis suggests that, while reasoning is overall effective, its potential is constrained by a hidden factor that acts as a significant bottleneck. To further examine this limitation, we perform a comprehensive error analysis. In particular, we classify incorrect responses of Vision-R1 under full reasoning into three categories, based on the characteristics of their reasoning trajectories:

1) **Perception Error**. Reasoning demonstrates notable deficiencies in the understanding and recognition of visual details such as chart misreading or hallucination exemplified in Fig. 1.

2) **Logical Error**. Models exhibit symbolic reasoning errors such as arithmetic inaccuracies, logically invalid inferences, or incoherence across intermediate reasoning steps.

3) **Knowledge Error**. There are commonsense violations or factual errors that may stem from outdated knowledge and inherent limitations in the pre-training phase.

We employ GPT-5 (Singh et al., 2025) to identify error categories and quantify their distribution, with details provided in Appendix A.3. Beyond this, we further investigate the impact of reasoning

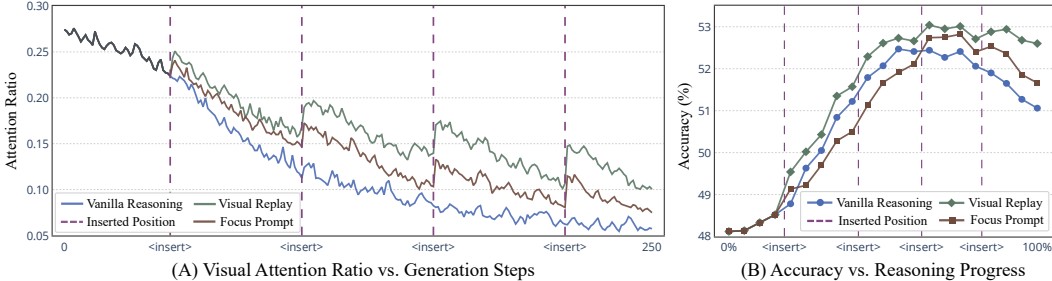

(A) Visual Attention Ratio vs. Generation Steps     (B) Accuracy vs. Reasoning Progress

Figure 3: The analysis of visual forgetting. In (A), we randomly select an example from MathVerse and visualize the evolution of attention ratio to visual tokens from Vision-R1 across generation steps. We consider vanilla reasoning as well as two variants, *i.e.*, visual replay and focus prompt. In (B), we report the average accuracy across established benchmarks and models, and track its trend throughout the reasoning process under the aforementioned three settings. The average cutoff positions across examples to insert images or instructions are indicated in vertical dash lines.

progress on these failure cases. Specifically, we apply early decision, where an error case is considered recoverable if the model can produce a correct answer at any earlier point along the reasoning trajectory, and report the corresponding proportion of such cases. These recoverable instances suggest that the model is initially on the right path but is ultimately misled by prolonged reasoning.

**The harder the model thinks, the worse the model sees**. As shown in Fig. 2 (B), despite the complexity of these reasoning benchmarks, the majority of errors made by VLMs stem not from logical failures (33.05%), but rather from basic perception errors (55.23%). Remarkably, a substantial portion of these perceptual errors (32.35%) may be recovered by terminating the reasoning process in advance through early decision. This counterintuitive result suggests that as reasoning progresses, the model's perceptual capability gradually weakens. Together with the finding in Fig. 2 (A), these degradation in visual perception caused by prolonged thoughts is most likely a key contributor to the performance decline observed in the later stages of reasoning.

**The harms of reasoning are most evident in vision-heavy tasks**. In Fig. 2 (C), we further examine the impact of reasoning progression on perception errors across different benchmarks. Notably, in MMStar and HallusionBench, which feature high-resolution real-world images and perceptually elusive content, the proportion of perception errors arising during reasoning is substantially higher. Moreover, the share of recoverable instances via early decision in these benchmarks is markedly greater than that observed in others, such as MathVerse and LogicVista. This observation suggests that, in contrast to tasks with simple visual structures, *e.g.*, geometry, the shortcomings of reasoning become more pronounced when applied to vision-intensive problems.

## 3.2 SIDE EFFECT OF REASONING: VISUAL FORGETTING

The above findings highlight a severe trade-off between logical capability and perceptual grounding in multimodal reasoning, which has emerged as a major bottleneck in the development of VLMs. Therefore, to further probe the underlying causes, we track the evolution of attention ratio assigned to visual tokens at each generation step, which provides a proxy of how visual information contributes to the reasoning process. As shown in Fig. 3 (A), under vanilla reasoning, the attention of visual tokens exhibits a marked decline as reasoning progresses, eventually reaching negligible levels. This pattern indicates the model's decision-making becomes increasingly driven by its own historical thoughts rather than the visual cues, a phenomenon termed visual forgetting. To validate this hypothesis, we design two straightforward inference-level remedies:

1) **Visual Replay**. Instead of presenting the visual input only at the beginning, we reintroduce the image to models at regular intervals throughout the reasoning process.

2) **Focus Prompt**. Similarly, at regular intervals, we explicitly prompt models to revisit the input image with instructions such as "I need to see the image" or "I have to look back".

**Encouraging models to look more often boosts reasoning**. Back to Fig. 3 (A), we insert either the image or instruction at four positions along the reasoning trajectory, which are approximately evenly spaced and aligned with logical boundaries; see Appendix A.5 for more details. Notably, both approaches trigger a sharp and immediate increase in visual attention upon insertion. As shown in

Fig. 3 (B), this reinforcement of visual information alleviates the performance degradation observed during the reasoning process, with visual replay yielding particularly pronounced gains, ultimately outperforming vanilla reasoning by around $1.5\%$. This also empirically shows that visual forgetting is the fundamental cause preventing reasoning from realizing its full potential in multimodal tasks.

## 4 THE PROPOSED METHOD

### 4.1 PRELIMINARY

To introduce our method, we first revisit the standard RL-based algorithm, typically Group Relative Policy Optimization (GRPO), which is a simplified variant of Proximal Policy Optimization (PPO) that eliminates the critic model. In the multimodal context, consider a dataset $\mathcal{D}$, where each instance consists of a question $q$, a ground truth answer $y$, and a visual input $\mathcal{I}$. The objective of GRPO is to encourage high-quality responses while penalizing inferior ones by comparing a set of generated candidates. Formally, given a policy model $\pi_\theta$, we aim to maximize the following:

$$\mathcal{J}_{\text{GRPO}}(\theta) = \mathbb{E}_{(q,y,\mathcal{I})\sim\mathcal{D},\{o_i\}_{i=1}^{G}\sim\pi_{\theta_{\text{old}}}(\cdot|q,\mathcal{I})}$$

$$\left[\frac{1}{G}\sum_{i=1}^{G}\frac{1}{|o_i|}\sum_{t=1}^{|o_i|}\left(\min\left(r_{i,t}(\theta)\hat{A}_{i,t},\text{clip}\left(r_{i,t}(\theta),1-\epsilon,1+\epsilon\right)\hat{A}_{i,t}\right)-\lambda D_{\text{KL}}\left(\pi_\theta\,||\,\pi_{\text{ref}}\right)\right)\right]$$

$$\text{with } r_{i,t}(\theta) = \frac{\pi_\theta(o_{i,t}\mid q,\mathcal{I},o_{i,<t})}{\pi_{\theta_{\text{old}}}(o_{i,t}\mid q,\mathcal{I},o_{i,<t})}. \tag{1}$$

$G$ denotes the number of sequences $o_i$ sampled from the old policy $\pi_{\text{old}}$. $\epsilon$ and $\lambda$ are hyperparameters that constrain the clipping bounds to ensure on-policy training and penalize deviations from the reference policy $\pi_{\text{ref}}$ to stabilize optimization. The estimated token-level advantage $\hat{A}_{i,t}$ is derived by broadcasting the normalized sequence-level reward $R_i$, which is defined as follows:

$$\hat{A}_{i,t} = \frac{R_i - \text{mean}(\mathbf{R})}{\text{std}(\mathbf{R})}, \quad i = 1,\cdots,G, \tag{2}$$

where $\mathbf{R} = \{R_1, R_2, ..., R_G\}$ indicates the reward of the sequence group. The design of reward functions offers considerable flexibility. A widely adopted approach involves the use of verifiable rewards, which evaluate model responses by comparing them to ground-truth answers to derive accuracy-based and format-consistency signals.

### 4.2 VISION-ANCHORED POLICY OPTIMIZATION

Although GRPO has become a common practice for training reasoning models, our findings in Section 3 reveal that existing VLMs suffer from visual forgetting, constituting a major bottleneck in multimodal reasoning. To address this, we propose VISION-ANCHORED POLICY OPTIMIZATION (VAPO), a simple yet effective approach as a multimodal replacement for GRPO that steers the reasoning process toward visually grounded trajectories. As shown in Fig. 4, VAPO first generates a set of correct and incorrect claims that describe the visual details in the images. These claims are strategically inserted as visual anchors along the reasoning path, where the model is prompted to judge their veracity, thereby enabling ongoing evaluation of model's perceptual capability throughout the reasoning process. Consequently, we introduce a perception reward, aggregated over all anchor points, which explicitly incentivizes the model to retain and utilize visual cues during reasoning.

**Generating Visual Claims**. We use visual claims as a proxy to evaluate the model's perceptual capability at various reasoning stages. To serve this purpose, the claims need to satisfy two criteria: 1) balanced, *i.e.*, the set should contain an equal number of correct and wrong claims to avoid biased evaluation and 2) independent, *i.e.*, the model judgment of each claim must rely solely on visual input, rather than historical reasoning. Specifically, for each example, we employ GPT-5 to generate a diverse set of claims with balanced correctness. To enforce independence from reasoning contexts, we only provide visual input without revealing the corresponding question, thereby ensuring claim verification is grounded purely in perceptual understanding. See Appendix A.6 for more details.

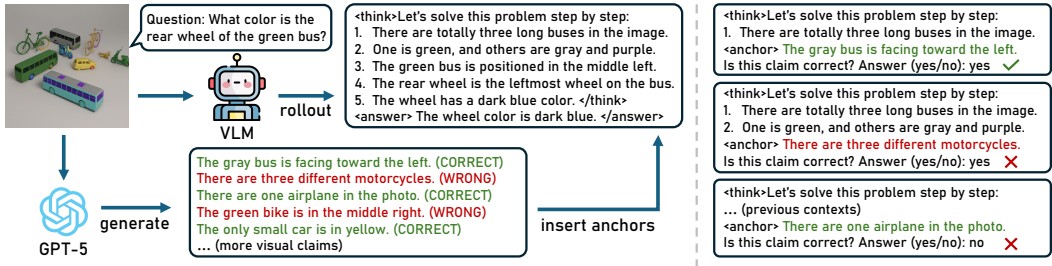

Figure 4: The overview of VAPO. On the left, we first employ GPT to generate a set of claims about visual input, each of which may be either factually correct or not. These claims are then treated as anchors and inserted into the model's reasoning process. Specifically, on the right, for each anchor, we randomly sample a prefix from the reasoning trajectory, append the claim to this truncated context, and probe the model judgment regarding the claim's validity.

**Inserting Visual Anchors**. Upon obtaining these claims, we set up a series of visual anchors within the model's reasoning process, functioning as intermediate checkpoints. Once reaching the anchor, the model is queried with a selected claim to assess its perceptual capability at that stage of reasoning. Formally, given a trajectory $o_i = (o_{i,1}, o_{i,2}, \cdots, o_{i,T})$, we define a set of anchors $\mathcal{A}_i = \{a_1, a_2, \cdots, a_K\}$ with $a_k \in [1, T]$ denoting the position randomly distributed throughout the trajectory. At each anchor $a_k$, a claim $c_k \in \mathcal{C}_i$ is sampled from the corresponding claim pool and appended to the prefix reasoning context, yielding a binary score over the judgement:

$$s_k = \mathbf{1}\left[\arg\max_{j \in \{\text{yes,no}\}} \pi_\theta(j \mid q, \mathcal{I}, o_{i,<a_k}, c_k) = l_k\right],\tag{3}$$

where $l_k$ denotes the ground truth of the claim. In other words, we evaluate the model's perceptual capability by probing its binary decision on the claim and scoring it against the reference label.

**Perception Reward**. Building on the anchors above, we introduce a perception reward that quantifies the model's overall perceptual capability throughout the reasoning process. Formally, given the per-anchor scores $\{s_k\}_{k=1}^K$, we design a late-emphasis weighted aggregation:

$$R_{\text{perc}} = \frac{\sum_{k=1}^K w_k s_k}{\sum_{k=1}^K w_k} \quad \text{with} \quad w_k = \exp\left(\beta \cdot \frac{a_k}{T}\right),\tag{4}$$

where $w_k$ indicates the weight assigned to the corresponding anchor, and $\beta$ is a hyperparameter that controls the degree of emphasis of later anchors. This design is motivated by the observation that the model's perceptual capability tends to decline as reasoning progresses. Consequently, greater weight is assigned to anchors in the latter stages to precisely target the model weakness. In summary, the final reward for a sampled sequence $o_i$ is computed as:

$$R_i = R_{\text{acc}} + R_{\text{fmt}} + \gamma \cdot \mathbf{1}\left[R_{\text{acc}} = 1\right] \cdot R_{\text{perc}},\tag{5}$$

where $R_{\text{acc}}$ and $R_{\text{fmt}}$ denote accuracy and format reward, $\gamma$ is the weight of perception reward. In practice, we impose an accuracy condition on perception reward to guard against reward hacking, thereby preventing models from simply boosting perceptual capability by producing trivially short reasoning paths. Our training procedure follows the objective of GRPO as defined in Section 4.1.

## 5 EXPERIMENT AND RESULTS

**Implementation Details**. Unless otherwise specified, all models are trained on ViRL39K (Wang et al., 2025a), a high-quality and comprehensive dataset, for 2 epochs with a learning rate of $5\mathrm{e}^{-6}$. We adopt Qwen2.5-VL (Bai et al., 2025) with 3B and 7B parameters as base models. For group reward computation, we follow previous works (Huang et al., 2025; Chen et al., 2025) by generating 5 responses per example with a sampling temperature of $1.0$. For VAPO, we set the default number of anchors to $K = 20$, the late-emphasis weight to $\beta = 1.5$, and reward weight $\gamma = 0.1$.

**Benchmarks**. We denote the models trained with our method as VAPO-Thinker, and evaluate on ten benchmarks covering diverse task types: mathmatical benchmarks such as MathVerse (Zhang

| Models | MathVerse | MathVista | MathVision | LogicVista | WeMath | Geometry3k | Avg. |
|---|---|---|---|---|---|---|---|
| *Close-source models* | | | | | | | |
| GPT-5-Thinking* | 81.2 | 81.9 | 72.0 | 70.0 | 71.1 | 79.9 | 76.1 |
| Gemini-2.5-Pro* | 76.9 | 80.9 | 69.1 | 73.8 | 78.0 | 77.2 | 75.9 |
| *Open-source models* | | | | | | | |
| Qwen2.5-VL-7B | 40.7 | 62.3 | 23.2 | 42.6 | 33.1 | 38.5 | 40.1 |
| InternVL2.5-8B* | 34.5 | 68.2 | 25.6 | 38.3 | 38.6 | 44.8 | 41.7 |
| R1-OneVision-7B | 46.4 | 64.1 | 29.9 | 45.6 | **44.6** | 46.1 | 46.1 |
| VLAA-Thinker-7B | 48.2 | 68.0 | 26.4 | 48.5 | 41.5 | 50.6 | 47.2 |
| Vision-R1-7B | 52.4 | 73.5 | 28.2 | 49.7 | 41.6 | 49.0 | 49.1 |
| *Our models* | | | | | | | |
| VAPO-Thinker-3B | 35.8 | 67.1 | 23.9 | 39.7 | 35.4 | 44.2 | 41.0 |
| VAPO-Thinker-7B | **53.3** | **75.6** | **31.9** | **50.9** | 43.6 | **51.3** | **51.1** |

Table 1: The evaluation of our proposed method on various mathematical benchmarks. We report results of existing open-source multi-modal reasoning models, as well as proprietary models for reference. Note that * indicates baseline results referenced from the OpenCompass leaderboard.

| Models | MMMU | MMStar | Hall | MMVet | Avg. |
|---|---|---|---|---|---|
| Qwen-VL | 52.7 | 54.9 | 50.0 | 64.8 | 55.6 |
| R1-OV | 54.3 | 54.1 | 52.5 | 65.2 | 56.5 |
| VLAA | 59.1 | 49.7 | 54.7 | 70.0 | 58.4 |
| V-R1 | 57.6 | 61.4 | 49.5 | 71.1 | 59.9 |
| VAPO | **60.2** | **63.0** | **57.4** | 71.9 | **63.1** |

| Models | WeMath | Geo3k | MMStar | Hall | Avg. |
|---|---|---|---|---|---|
| VLAA$_{FP}$ | 41.8 | 50.7 | 51.1 | 55.2 | 49.7 |
| VLAA$_{VR}$ | 42.3 | 51.1 | 52.9 | 56.2 | 50.6 |
| V-R1$_{FP}$ | 42.1 | 49.7 | 61.8 | 50.5 | 51.0 |
| V-R1$_{VR}$ | 42.5 | 50.5 | 62.1 | 51.8 | 51.7 |
| VAPO | **43.6** | **51.3** | **63.0** | **57.4** | **53.8** |

Table 2: The evaluation of our proposed method on general-purpose benchmarks. All baselines considered in this evaluation are of the 7B scale.

Table 3: Comparison with strong baselines augmented with test-time remedies. FP and VR denote focus prompt and visual replay, respectively.

et al., 2024), MathVista (Lu et al., 2023), MathVision (Wang et al., 2024b), LogicVista (Xiao et al., 2024), WeMath (Qiao et al., 2024) and Geometry3k (Lu et al., 2021), as well as general-purpose ones including MMMU (Yue et al., 2024), MMStar (Chen et al., 2024b), HallusionBench (Guan et al., 2024) and MMVet (Yu et al., 2023). Further evaluation results are provided in Appendix B.

**Baseline Methods**. To verify the effectiveness of our approach, we compare VAPO-Thinker with a range of existing strong reasoning models, including proprietary ones such as GPT-5-Thinking (Singh et al., 2025) and Gemini-2.5-Pro (Comanici et al., 2025), as well as open-source counterparts encompassing InternVL2.5 (Chen et al., 2024d), R1-OneVision (Yang et al., 2025c), VLAA-Thinker (Chen et al., 2025), and Vision-R1 (Huang et al., 2025). The baseline results are primarily referenced from the corresponding papers, secondarily from the OpenCompass leaderboard, and reproduced when neither source is available. See Appendix A.7 for more configuration details.

## 5.1 MAIN RESULTS

**VAPO consistently improves accuracy across diverse benchmarks**. As shown in Table 1, VAPO-Thinker-7B outperforms recent reasoning models of the same scale on mathematical problems, achieving an average improvement of 2% (49.1% → 51.1%). The advantage is more pronounced on general-purpose tasks, as shown in Table 2, where our method surpasses previous best results by 3.2% (59.9% → 63.1%), thereby establishing a new state of the art. This suggests that compared to logic-heavy problems such as math, VAPO is particularly effective on vision-intensive tasks like MMStar and HallusionBench, demonstrating stronger visually grounded reasoning.

**VAPO offers a principled solution beyond inference-level remedies**. In Section 3.2, we introduce two remedies, *i.e.*, visual replay and focus prompt, as preliminary exploration to verify the visual forgetting issue. Here we compare VAPO against these test-time strategies. As shown in Table 3, even when strong baselines are equipped with visual replay or focus prompt, VAPO still achieves substantially higher performance, suggesting that inference-level remedies alone are insufficient to address visual forgetting, whereas VAPO fundamentally rectifies this reasoning deficiency.

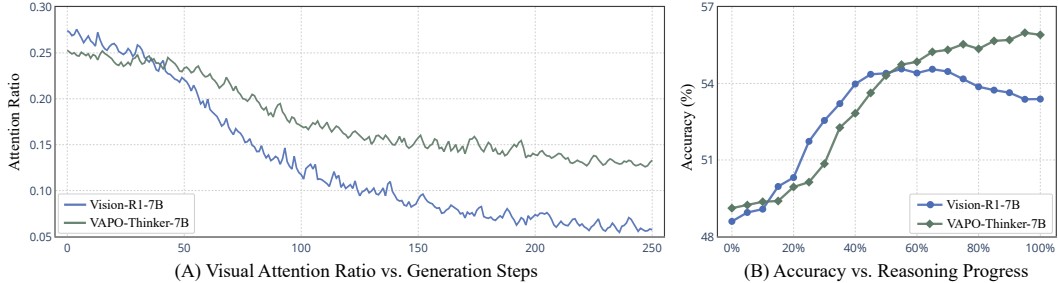

Figure 5: Zoom-in analysis of our method. In (A), we compare the evolution of visual attention across generation steps between VAPO and the baseline on the selected example. In (B), we track the average accuracy across ten benchmarks throughout the reasoning process via early decision.

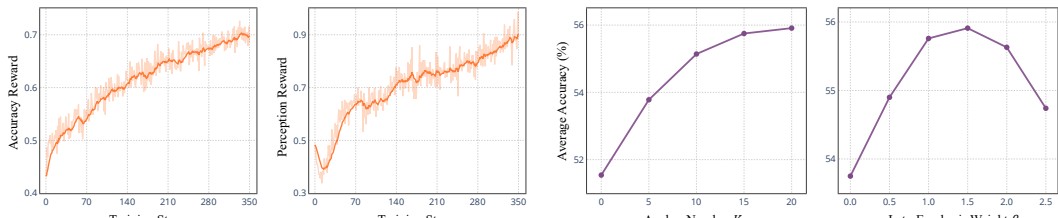

Figure 6: The learning curves of accuracy and perception reward during the training stage.

Figure 7: The ablation study on the effects of anchor number $K$ and late-emphasis weight $\beta$.

## 5.2 FURTHER DISCUSSION

**VAPO fully releases the potentials of reasoning**. Here we analyze the evolution of visual attention and accuracy over the course of reasoning process. In Fig. 5 (A), compared with the baseline, our model demonstrates a more gentle decline in the attention ratio, sustaining a consistently higher level in the later stages, indicating that VAPO effectively strengthens the contribution of visual cues to model decisions. The benefit brought by this is directly reflected in accuracy, as shown in Fig. 5 (B), where in contrast to the baseline which exhibits a sharp performance decline during later steps, our method achieves steadily increasing accuracy, thereby fully leveraging the advantages of reasoning.

**VAPO greatly enhances perceptual capability during reasoning**. In Fig. 6, we plot the training curves of accuracy and perception reward. The results reveal that perception reward exhibits a steady upward trend in parallel with accuracy, suggesting that the model progressively allocates greater attention to visual input and thereby improves its ability to verify visual claims. Interestingly, an sharp decline in perception reward is observed during the early stage, which may be attributed to the initial dominance of accuracy and format alignment, causing the model to temporarily disregard visual information. This underscores a fundamental limitation of prior standard training paradigms.

## 5.3 ABLATION STUDY

**More anchors yield better visually grounded reasoning**. We examine the impact of visual anchor number $K$ on average accuracy across ten benchmarks. As shown on the left of Fig. 7, we vary $K$ from 0, which degrades to vanilla GRPO training, to 20, which is adopted as our default configuration. The results indicate that the average accuracy improves rapidly as $K$ increases and approaches saturation at around $K = 20$. This behavior is intuitive, as setting up denser anchors during reasoning provides a more reliable assessment of the model's perceptual capability.

**Later anchors contribute to greater gains**. In VAPO, the parameter $\beta$ controls the relative emphasis placed on later anchors to target longer reasoning. As shown on the right of Fig. 7, we evaluate the effect of varying $\beta$ on average accuracy. When $\beta = 0$, all anchors are assigned equal weights, while larger values assign greater importance to later anchors. The results show that performance peaks at $\beta = 1.5$, where around 50% of the total weight is concentrated on the last 30% of anchors. More results such as ablation of reward weight $\gamma$ and limitation analysis are provided in Appendix B.

## 6    CONCLUSION

In this study, we conduct an investigation into the capabilities and limitations of multimodal reasoning. We first reveal the dual nature of reasoning: while it enhances logical inference and proves beneficial for challenging problems, longer reasoning length may gradually impair perceptual grounding, leading to recognition deficiencies on otherwise basic visual questions. Then we further identify this phenomenon and attribute it to visual forgetting, where prolonged reasoning length causes the model to disregard visual information. To address this, we propose VISION-ANCHORED POLICY OPTIMIZATION (VAPO), a simple yet effective method that steers the reasoning process along trajectories anchored in visual evidence. Our result model, `VAPO-Thinker-7B`, effectively mitigates visual forgetting and establishes a new state of the art across various benchmarks.

### ACKNOWLEDGEMENT

This research was, in part, funded by the U.S. Government – DARPA TIAMAT HR00112490421. The views and conclusions contained in this document are those of the authors and should not be interpreted as representing the official policies, either expressed or implied, of the U.S. Government. We also gratefully acknowledge Lambda GPU Cloud and Maincode for their generous provision of computational resources.

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

# CONTENTS

# A EXPERIMENTAL DETAILS

## A.1 BENCHMARK STATISTICS

In the main paper, we provide a high-level overview of the selected benchmarks. Here, we present detailed information for clarity and ease of reproducibility, including dataset types, sizes, and splits. All evaluation scripts are implemented using the VLMEvalKit framework (Duan et al., 2024).

1) **MathVerse** is a benchmark comprising 2,612 high-quality mathematical questions. Each example provides varying levels of multimodal information. Following prior work, we evaluate on the Test Mini split using the Vision Only setting, which includes approximately 700 samples.

2) **MathVista** is a comprehensive mathematical benchmark that evaluates a range of skills including puzzle solving, algebraic reasoning, and scientific understanding, and comprises 6,141 examples. For our evaluation, we use the Test Mini split, which contains approximately 1,000 examples.

3) **MathVision** comprises 3,040 high-quality mathematical problems sourced from real-world math competitions, spanning 16 distinct disciplines and five levels of difficulty, providing a comprehensive and challenging benchmark for evaluating VLMs. We conduct our evaluation using the full test set.

4) **LogicVista** assesses the fundamental logical reasoning capabilities of VLMs, covering a range of reasoning types including spatial, deductive, inductive, numeric, and mechanical. The benchmark comprises 448 visual multiple-choice questions. We conduct our evaluation on the full test set.

5) **WeMath** comprises 6.5k visual math questions structured around 67 hierarchical knowledge concepts across five levels of granularity. We evaluate on the Test Mini split, which contains approximately 1,740 examples, and report the strict score as the primary evaluation metric.

6) **Geometry3k** consists of 3,002 geometry problems with dense annotations in formal language, requiring abstract problem-solving and symbolic reasoning based on axiomatic knowledge. For evaluation, we combine the validation and test splits, resulting in approximately 900 examples.

7) **MMMU** is a challenging benchmark that covers a broad range of disciplines, requiring college-level subject knowledge and reasoning. It contains 11.5k curated multimodal questions sourced from college exams, quizzes, and textbooks. We perform our evaluation on the validation split.

8) **MMStar** is a vision-indispensable multimodal benchmark specifically designed to ensure that each sample exhibits strong visual dependency and requires advanced multimodal reasoning capabilities. It comprises 1,500 samples for offline evaluation. Here we evaluate on the full test set.

9) **HallusionBench** is designed to challenge advanced VLMs by emphasizing fine-grained understanding and interpretation of visual information. It consists of 346 manually curated images paired with 1,129 questions. In this study, we conduct our evaluation on the full test split.

10) **MMVet** comprises 218 examples and defines six core vision-language capabilities, focusing on their integration to evaluate the synergy among different skills. It is designed to assess the overall competence of generalist models. We perform our evaluation on the full test split.

## A.2 BASELINE SETTINGS

In the main text, we compare our method against several popular baselines to validate its effectiveness. For models such as GPT-5-Thinking, Gemini-2.5-Pro, and InternVL-2.5, we directly report results from the OpenCompass leaderboard. For other models, we provide reproduced results when official numbers are not available in the original papers. Below, we detail the settings used for these reproduced baseline models for reference. All results are reproduced using greedy decoding.

1) **R1-OneVision** includes both 3B and 7B variants, which are first trained via SFT followed by RL, with a particular focus on mathematical reasoning tasks. In our experiments, we adopt the publicly available 7B checkpoint based on Qwen2.5-VL-7B.

2) **VLAA-Thinker** is an RL-only model trained on a high-quality and challenging dataset, and is the first to demonstrate that RL outperforms SFT in multimodal settings. In our experiments, we use the VLAA-Thinker-7B variant, which is also based on Qwen2.5-VL-7B.

ROLE
You are an impartial evaluator. Given an image, a question, the ground truth answer, and the model's reasoning traces, identify the PRIMARY error category in the reasoning process.

ERROR TYPE (CHOOSE ONE PRIMARY)
A) Perception Error: Failure to correctly perceive, localize, or read visual content (e.g., chart misreading, counting mistakes, OCR/label misread, visual hallucination/attribution errors).

B) Logical Error: Symbolic/algorithmic reasoning mistakes given correctly perceived facts (e.g., arithmetic slip, invalid inference, algebraic misstep, inconsistent intermediate steps).

C) Knowledge Error: Missing/incorrect factual or commonsense/world knowledge not visually available (or clearly contradicted by known facts), including outdated knowledge or misremembered facts.

D) Others: No observable error in reasoning, or the reasoning content is too short or uninterpretable.

OUTPUT
Return exactly one character: A, B, C or D. Do not include any other text, spaces, punctuation, or quotes.

Figure 8: The designed prompt to instruct GPT-5 to assist in categorizing the different error types.

3) **Vision-R1** is a powerful reasoning model trained at scale, with the combined amount of RL and SFT data being approximately five times larger than that used in our setting. In our experiments, we use Vision-R1-7B, which represents the strongest variant reported by the authors.

## A.3    ERROR ANALYSIS

In Section 3.1, we conduct a comprehensive error analysis by leveraging GPT-5 to categorize different types of errors and quantify their distributions, aiming to assess the impact of reasoning on failure cases. The prompt used for GPT-5 input is shown in Fig. 8. As illustrated, we provide GPT-5 with four options: three predefined error categories, along with an Others category to account for a small number of outliers, such as cases with uninterpretable content. We then compute the proportion of each error type to support our analysis. For completeness, in addition to the perception error examples as shown in Fig. 1, here we also provide representative cases of logical errors and knowledge errors, as illustrated in Fig. 9.

In addition, based on the recorded human audit of 500 randomly sampled failure cases, we find that instances containing multiple error types within the same response are quite rare, occurring in fewer

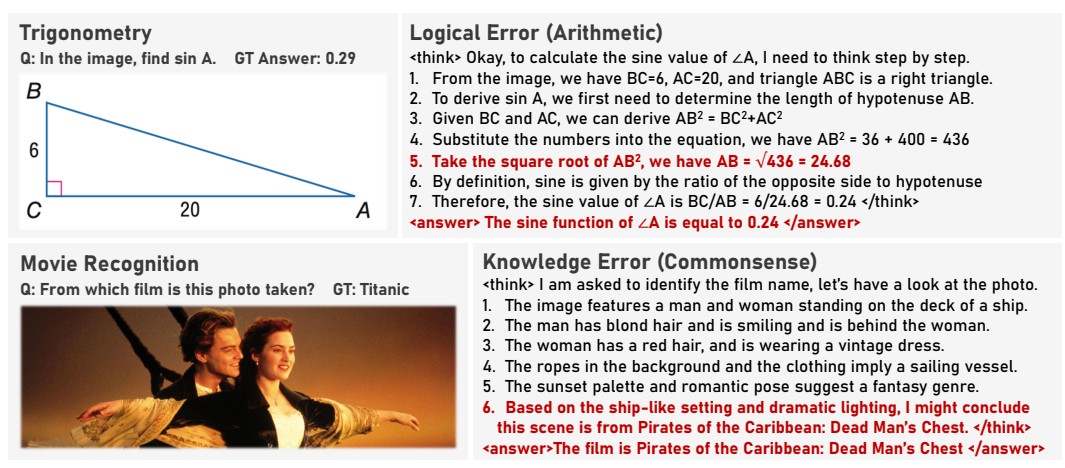

Figure 9: The representative examples of logical and knowledge errors.

| Error Category | Logic → Perc | Perc → Logic | Knowledge → Perc | Knowledge → Logic |
|:---:|:---:|:---:|:---:|:---:|
| # Examples | 5 | 7 | 3 | 1 |

Table 4: The number of examples with multiple error categories co-occuring.

than 20 examples ($< 4\%$), which is also the reason we prompt GPT to directly identify the primary error category as shown in Fig. 8. To further investigate this, we manually re-examine these cases and record the specific ordering patterns in which different error categories co-occur. As shown in Table 4, any error category combinations not listed above did not occur in our samples. Notably, cases in which logical errors precede perceptual errors are extremely rare. This rarity ensures that such patterns do not meaningfully affect our experimental results or conclusions. It also makes it difficult to draw any substantive connection between perceptual errors and other error types, given how infrequently they co-occur in practice. These results further reinforce our finding that longer reasoning contexts tend to induce visual forgetting, which in turn leads to the observed performance degradation.

## A.4   ATTENTION VISUALIZATION

In the main paper, we visualize the attention over image tokens to reflect the contribution of visual information to the model's reasoning process. Here, we provide additional details on the implementation. For each generation step, we compute the sum of attention scores after softmax assigned by the output token to all preceding image tokens. This yields a ratio between 0 and 1, which is then averaged across all layers to produce the final attention value. Additionally, in Fig. 3 (A), since inserting images, *i.e.*, visual replay, or instructions, *i.e.*, focus prompt, may alter the final reasoning trajectory and result in slightly different sequence lengths, we normalize the comparison by truncating all outputs to the first 250 tokens, which basically covers the full response. This strategy is also applied in Fig. 5 (A) when comparing the visual attention between Vision-R1 and `VAPO-Thinker`.

---

ROLE
You are given an image. Generate EXACTLY 20 DISTINCT visual claims about THIS image.

DEFINITION: A visual claim is a short, factual statement about something that is CLEARLY VISIBLE in the image (no guesses, no context outside the image).

HARD CONSTRAINTS:
• Each claim should be no more than 20 words.
• Each claim must be obviously verifiable from the pixels.
• One fact per line (no conjunctions; no explanations).
• English only; avoid hedges ("maybe", "appears", "likely").

CORRECTNESS:
• Produce 10 CORRECT and 10 WRONG claims (total = 20).
• Wrong claims must be plausible and possible perception errors
• Do not make logically impossible statements (e.g., "left of itself").
• Correct and wrong claims should be independent, not just simple opposites of each other.
• Label each correct claim with [CORRECT] at the end of the line.
• Label each wrong claim with [WRONG] at the end of the line.

OUTPUT FORMAT (MUST MATCH EXACTLY):
• Return ONLY the 20 numbered lines, nothing else.
• Numbering style: "1. …" through "20. …"
• One line per claim.

Figure 10: The user prompt for GPT-5 to generate visual claims for `VAPO`.

## A.5 INFERENCE-LEVEL REMEDIES

In the main text, we introduce two inference-level remedies, *i.e.*, visual replay and focus prompt, as preliminary strategies to demonstrate the negative impact of visual forgetting on reasoning. For completeness, we provide a detailed description of both approaches here.

In the visual replay strategy, the input image is reintroduced periodically during the model's reasoning process at regular intervals. In practice, to improve efficiency and prevent exceeding the model's context length, the reinserted image is downsampled to a lower resolution. Furthermore, the insertion points are selected to satisfy two criteria: (1) uniform segmentation of the reasoning trajectory, and (2) alignment with logical boundaries such as commas, periods, or line breaks to avoid interrupting syntactic or semantic units.

For the focus prompt strategy, at each insertion point, we randomly sample one prompt from a set of three manually designed instructions including "I need to see the image", "I have to look back" and "Let me verify against the visual input", to ensure robustness against prompt variability. For alignment, we adopt the nearby insertion positions as used in visual replay, facilitating a consistent comparison between the two methods. It is worth noting that in visual replay, the image is reintroduced as part of the user prompt in a dialogue format, whereas in focus prompt, the instruction is directly injected into the assistant's response.

## A.6 VISUAL CLAIM GENERATION

Our method encourages the model to rely more effectively on visual input by introducing a perception reward, which is derived from the model's ability to evaluate a set of visual claims during the reasoning process. This evaluation serves as a proxy for assessing the model's perceptual capability at varying reasoning stages. To support this, we prompt GPT-5 to generate a specified number of visual claims conditioned on the input image. An example of the user prompt used for this purpose is shown in Fig. 10. We require the generated visual claims to be concise factual statements, free from hedging terms such as possibly or appears. Each set of claims must contain an equal number of correct and incorrect statements. Furthermore, the claims are constructed to be independent of the corresponding example question, ensuring that the model must refer to the visual input rather than relying on prior outputs when evaluating their validity. While several generated visual claims have already been presented in Fig. 4, we provide more illustrative examples in Fig. 11 for reference.

In addition, we employ a rule-based filtering mechanism and dynamically generate visual claims. That is, for each training example, any claim that fails to meet the required criteria is discarded and resampled until the target number of valid claims is reached. The filtering process incorporates

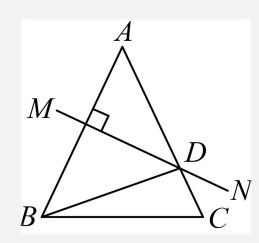
1. Segment BD is parallel to the segment BC. [WRONG]
2. BC is a vertical line segment in the triangle. [WRONG]
3. Point C is at the bottom-right of the triangle. [CORRECT]
4. The label M is on the left outside the triangle. [CORRECT]
5. The letter A is at the top vertex of the triangle. [CORRECT]
6. The right-angle square marks the angle at vertex B. [WRONG]
…

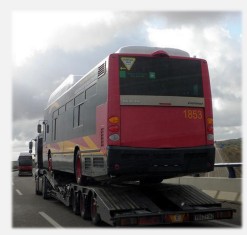
1. Many bus windows are shattered. [WRONG]
2. The bus sits on a black flatbed trailer. [CORRECT]
3. The direction of the bus faces the camera. [WRONG]
4. The scene is captured on a narrow city street. [WRONG]
5. The sky is filled with gray and thick clouds. [CORRECT]
6. The rear of the bus displays the number "1853". [CORRECT]
…

Figure 11: The examples of the GPT-generated visual claims.

multiple constraints, as detailed in the prompt specification in Fig. 10, including limits on claim length, the prohibition of ambiguous expressions (e.g., "maybe", "appears"), and the requirement to maintain a balanced ratio of correct and incorrect claims. To reduce redundancy, we compute pairwise embedding distances between generated claims using Qwen3-Embedding-0.6B (Zhang et al., 2025), and remove claims with excessively high semantic similarity ($> 0.95$). We allow straightforward low-level claims such as color naming, while we impose a mild diversity constraint that limits the number of purely low-level color or counting statements per image through regular expression, favoring a richer distribution of relational, text-based, and global scene-level claims.

## A.7 TRAINING CONFIGURATION

In Section 5, we briefly outline the training setup. Here, we provide a more detailed description. By default, we adopt ViRL39K as our training dataset, which is a refined collection derived from multiple existing sources such as MM-Eureka (Meng et al., 2025), MV-Math (Wang et al., 2025c), and M3CoT (Chen et al., 2024c). The dataset covers a wide spectrum of domains, including STEM subjects, social topics, chart reasoning, and spatial relations. It is worth noting that prior baselines rely on substantially larger datasets that combine SFT and RL, for example, 155k samples for R1-OneVision and 210k samples for Vision-R1. Although this comparison is not fair for us in terms of training data scale, it further highlights the effectiveness of our proposed method.

During the rollout phase, we sample 5 responses per example with a temperature of $1.0$, and employ vLLM as the backend to accelerate decoding. We then insert visual anchors into these generated responses to evaluate the model's perceptual capability. For each anchor, we randomly sample a prefix of the reasoning content, append a visual claim, and instruct the model to judge its correctness. To ensure binary decision-making, the decoding process is constrained to produce either yes or no, with the temperature fixed at $0.0$. Importantly, throughout this stage, we adopt the default system prompt of Qwen2.5-VL without explicitly introducing the presence or meaning of anchors. This design avoids altering the model's inherent behavior and keeps anchors fully transparent, serving as an implicit reward mechanism that encourages the model to rely more heavily on visual cues.

For training, we employ 8 NVIDIA A100-80G GPUs. The policy loss follows the default configuration of GRPO, where the clip ratio $\epsilon$ is set to $0.2$, and the KL penalty coefficient $\lambda$ is fixed at $1e^{-2}$. For VAPO, we set the anchor number $K = 20$, the late-emphasis weight $\beta = 1.5$, and the perception reward weight $\gamma = 0.1$. The entire training process is conducted using verl (Sheng et al., 2025).

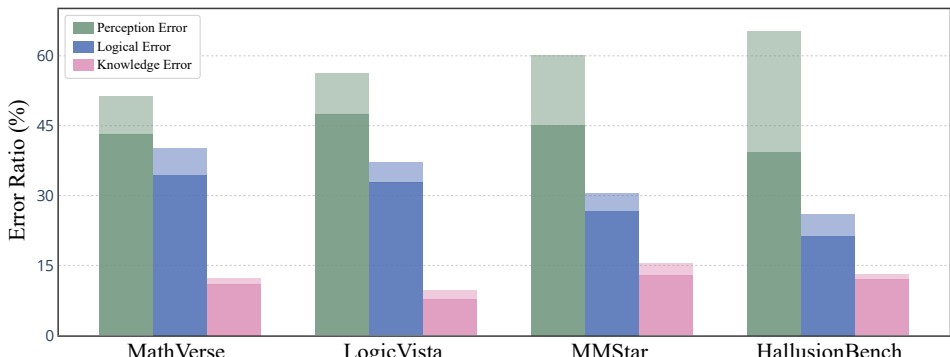

Figure 12: The error ratio of Vision-R1 as well as the correction rate achieved by our method across benchmarks. In the bar chart, the full height of each bar represents the overall error rate of Vision-R1, with the light-colored segment indicating the proportion of errors successfully corrected by our method, and the dark-colored segment corresponding to the remaining uncorrected errors.

| $\gamma$ | MathVerse | LogicVista | Geometry3k | MMMU | MMStar | HallBench | Avg. |
|---|---|---|---|---|---|---|---|
| 0.00 | 44.2 | 44.4 | 44.5 | 53.6 | 55.2 | 51.2 | 49.0 |
| 0.05 | 47.7 | 46.3 | **48.2** | 55.5 | 58.6 | 53.7 | 51.7 |
| 0.10 | **48.9** | 47.3 | 47.7 | **56.7** | 59.1 | 55.5 | **52.5** |
| 0.15 | 48.4 | **46.7** | 48.1 | 56.2 | **59.6** | **56.7** | 52.4 |
| 0.20 | 47.9 | 46.6 | 47.4 | 55.7 | 59.4 | 54.2 | 51.9 |

Table 5: The benchmark accuracy with varying perception reward weight $\gamma$.

## B    MORE EVALUATION

### B.1    ERROR CATEGORY DISTRIBUTION

In the main text, we analyze the error cases of the representative baseline Vision-R1 and observe that perception errors constitute the majority, underscoring the detrimental impact of reasoning on visual information utilization. To assess the effectiveness of our approach in addressing this issue, we further examine the correction rate of our method across the different error categories of the baseline model, as illustrated in Fig. 12. We observe that, for cases where the baseline model fails, our method substantially corrects a significant portion of perception errors, with improvements reaching up to 20% on vision-intensive benchmarks. This perceptual correction capability constitutes a major source of the performance gains achieved by our approach.

### B.2    VISUAL ANCHOR SCORE

Our method assesses the model's overall perceptual capability by setting up visual anchors at different stages of the reasoning process. As shown in Fig. 6, the perception reward exhibits a clear upward trend throughout training, indicating that the model becomes increasingly proficient at evaluating visual claims and progressively relies more on visual information. To further investigate the role of individual visual anchors, we visualize the distribution of anchor scores before and after training, *i.e.*, comparing the base model with the model after applying our method. As shown in Fig. 13, the base model, *i.e.*, Qwen2.5-VL-7B, exhibits a clear downward trend in anchor scores as reasoning progresses, eventually approaching the level of random guessing (around 0.5). In contrast, our trained model significantly improves anchor scores, maintaining a stable level around 0.9 throughout the reasoning process. This indicates a substantial enhancement in perceptual capability as a result of our method.

### B.3    PERCEPTION REWARD WEIGHT

In the main text, we have conducted ablation studies on the anchor number $K$ and the late-emphasis weight $\beta$ to identify their optimal settings. Here, we further investigate the impact of perception reward weight $\gamma$ on model performance. For computational efficiency, in this experiment we do not

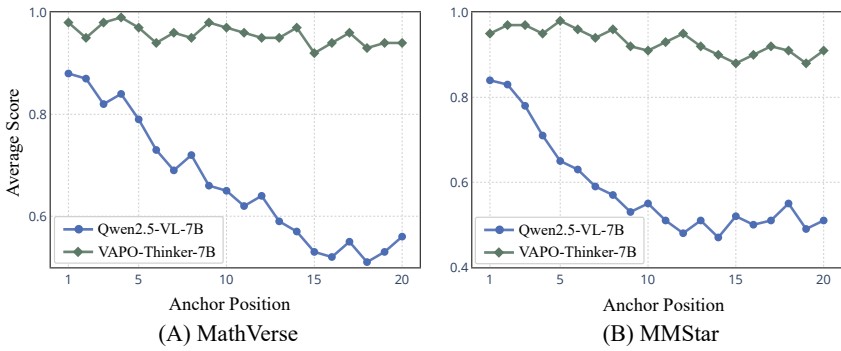

(A) MathVerse          (B) MMStar

Figure 13: The anchor score across varying anchor positions, where a higher index corresponds to a later stage in the reasoning process. Scores are averaged across all examples within the benchmark.

| Method | MathVerse | LogicVista | Geometry3k | MMMU | MMStar | HallBench | Avg. |
|---|---|---|---|---|---|---|---|
| GRPO | 44.2 | 44.4 | 44.5 | 53.6 | 55.2 | 52.2 | 49.0 |
| GRPO$_{aug}$ | 44.7 | 45.2 | 43.9 | 54.2 | 54.8 | 53.1 | 49.3 |
| VAPO | **45.9** | **46.1** | **46.7** | **55.2** | **57.8** | **54.3** | **51.0** |

Table 6: The comparison with VAPO and GRPO augmented with visual claim examples.

| Method | MathVerse | LogicVista | Geometry3k | MMMU | MMStar | HallBench | Avg. |
|---|---|---|---|---|---|---|---|
| GRPO | 44.2 | 44.4 | 44.5 | 53.6 | 55.2 | 51.2 | 49.0 |
| NonVisualClaim | 45.7 | 44.9 | 44.3 | 54.8 | 57.2 | 51.9 | 49.8 |
| QAlignedClaim | 44.9 | 45.8 | 46.1 | 55.3 | 57.8 | 53.5 | 50.6 |
| VisClaim | **48.9** | **47.3** | **47.7** | **56.7** | **59.1** | **55.5** | **52.5** |

Table 7: The comparison with non-visually-dependent claims and question-aligned claims.

train on the full dataset but instead randomly sample 5000 examples from the full 39k training set. As shown in Table 5, setting $\gamma = 0$ reduces the training procedure to vanilla GRPO without perceptual supervision. As $\gamma$ increases, the average accuracy across benchmarks improves significantly, peaking around $\gamma = 0.1$. Notably, mathematical tasks tend to favor smaller values of $\gamma$, which is intuitive given the relatively simple visual structures of these tasks. In contrast, vision-intensive benchmarks such as MMStar and HallucinationBench benefit from more aggressive settings, reflecting their greater reliance on strong perceptual capability.

### B.4    ABLATION ON DATA AUGMENTATION

In our approach, visual claims play a central role by providing reward signals that explicitly encourage the model to rely more heavily on visual inputs. This mechanism helps mitigate visual forgetting and improves accuracy across benchmarks. However, since these visual claims are generated by GPT and accompanied by correctness labels, they may be viewed as new synthetic examples augmented to the original training data. This raises a critical question: are the performance gains brought by VAPO primarily attributable to the reward-driven promotion of visually grounded reasoning, or are they simply a result of data augmentation through the inclusion of these synthetic visual examples?

To investigate this question, we introduce a new baseline in which the generated visual claims are directly transformed into additional training examples and integrated into the original dataset. For computational efficiency, we randomly sample 5000 examples from the full dataset, as described in Appendix B.3. For each example, we generate 5 visual claims, resulting in a fivefold expansion of the training set to 30k examples. This setup allows us to isolate the performance gains attributable purely to data augmentation with synthetic visual claims. The baseline is trained using the standard GRPO algorithm without any additional modifications. In contrast, under the VAPO framework, these same visual claims are used as anchors during training, with the anchor number set to $K = 5$ to match the available number of claims per instance.

As shown in Table 6, augmenting vanilla GRPO with additional examples constructed from visual claims yields only marginal improvement over the original dataset ($49.0 \rightarrow 49.3$). In contrast, our method achieves a substantial performance gain of $2\%$ ($49.0 \rightarrow 51.0$). This discrepancy may be attributed to two key factors: 1) The visual claims are generally short, simple, and strongly dependent on visual information, often solvable without long reasoning trajectories, thus offering limited learning capacity; 2) The augmented examples are derived from a small number of unique images, *i.e.*, five examples share the same image, significantly reducing data diversity and increasing the risk of overfitting. These results suggest that interpreting the role of visual claims purely as a form of data augmentation is not appropriate. Rather, the performance gains of VAPO are primarily driven by its promotion of visually grounded reasoning.

| $\lambda\,(1e^{-2})$ | MathVerse | LogicVista | Geometry3k | MMMU | MMStar | HallBench | Avg. |
|---|---|---|---|---|---|---|---|
| 0 | 48.1 | 47.1 | 47.3 | 57.1 | 59.3 | 55.3 | 52.4 |
| 1 | **48.9** | **47.3** | 47.7 | 56.7 | 59.1 | **55.5** | **52.5** |
| 2 | 48.3 | 47.0 | 47.9 | **57.2** | 58.4 | 54.9 | 52.3 |
| 5 | 48.1 | 46.2 | **48.3** | 57.1 | **59.8** | 54.3 | 52.3 |

Table 8: The impact of KL penalty coefficient $\lambda$ to the benchmark accuracy.

| Method | MathVerse | LogicVista | Geometry3k | MMMU | MMStar | HallBench | Avg. |
|---|---|---|---|---|---|---|---|
| VAPO | 53.3 | 50.9 | 51.3 | 60.2 | 63.0 | 57.4 | 56.0 |
| VAPO$_{FP}$ | 53.1 | 51.2 | 50.7 | 60.5 | 63.4 | 57.9 | 56.1 |
| VAPO$_{VR}$ | 53.7 | 50.4 | 51.8 | 57.8 | 62.7 | 58.2 | 55.8 |

Table 9: The impact of VAPO augmented with inference-level remedies.

## B.5 EFFECT OF VISUAL CLAIMS

In our method, visual claims serve as a critical proxy for measuring a model's perceptual capability. To better understand their influence, we investigate how different types of visual claims affect model performance. In addition to our default setting, we consider two alternative variants: 1) non-visually-dependent claims: although these are factual statements about the image, they emphasize external knowledge or logical reasoning rather than concrete visual details; 2) question-aligned claims: these are closely related to the specific question associated with the image, potentially allowing the model to assess claim correctness based on its own historical reasoning outputs rather than pure visual grounding. To generate these claims, we prompt GPT accordingly: for the first variant, we explicitly instruct GPT to produce claims that are not visually dependent; for the second, we provide both the image and the corresponding question, asking it to output claims relevant to the question content. For efficiency, we conduct the analysis using 5000 examples sampled from the full training set.

As shown in Table 7, both the non-visually-dependent and question-aligned variants perform substantially worse than our default visual claim setup. This is likely due to the fact that non-visually-dependent claims can often be judged correctly without requiring strong perceptual capability, thereby diminishing the core purpose of VAPO. On the other hand, question-aligned claims are highly correlated with the question content, allowing the model to infer their correctness from its own reasoning traces without relying on visual input. These results highlight that the design of visual claims is crucial to the effectiveness of the VAPO framework.

## B.6 KL PENALTY COEFFICIENT

In the previous sections, we have analyzed the impact of VAPO-specific hyperparameters, including the anchor number $K$, the late-emphasis weight $\beta$, and the perception reward weight $\gamma$. For the underlying GRPO framework, we adopt the default setting for the KL penalty coefficient $\lambda = 1e^{-2}$. For completeness, here we also examine the effect of varying $\lambda$ on the performance of VAPO. To maintain computational efficiency, we conduct this analysis on a randomly sampled subset of 5000 examples from the full training set. As shown in Table 8, VAPO exhibits minimal sensitivity to the choice of $\lambda$. Regardless of whether the KL penalty is applied or varied in magnitude, the average accuracy fluctuates by no more than $0.2\%$.

## B.7 VAPO WITH INFERENCE-LEVEL REMEDIES

To further compare the effectiveness of our method with inference-level remedies, *i.e.*, visual replay and focus prompt, we augment VAPO by incorporating these two straightforward approaches. We follow the same insertion strategy as in Fig. 3, where interventions are applied at four points throughout the reasoning process. As shown in Table 9, augmenting our model with inference-level remedies for visual forgetting results in minimal impact, with performance fluctuations within approximately $0.2\%$. This suggests that our method has already substantially enhanced the model's reliance on visual input, effectively subsuming the benefits provided by these additional strategies.

| Method | MathVerse | LogicVista | Geometry3k | MMMU | MMStar | HallBench | Avg. |
|---|---|---|---|---|---|---|---|
| GRPO | 48.2 | 45.5 | 47.3 | 56.6 | 58.9 | 53.2 | 51.6 |
| VAPO$_{attn}$ | 47.6 | 45.8 | 46.4 | 57.3 | 59.4 | 54.8 | 51.8 |
| VAPO$_{perc}$ | **53.3** | **50.9** | **51.3** | **60.2** | **63.0** | **57.4** | **56.0** |

Table 10: The comparison with attention reward which directly maximizes visual attention ratio.

| Method | Epoch | Time | Accuracy | Gain |
|---|---|---|---|---|
| GRPO | 2 | 18h46m | 51.54 | — |
| DAPO | 2 | 25h11m | 53.17 | +1.63 |
| VAPO | 2 | 19h14m | 55.91 | +4.37 |

Table 11: The time costs and gains for different policy gradient algorithms.

## B.8 ATTENTION-BASED REWARD

In the main text, we use the evolution of visual attention throughout the reasoning process as a proxy to examine whether our method mitigates visual forgetting. Therefore, a natural baseline to consider is a more trivial alternative: directly using the attention scores of image tokens as a reward signal to guide training. To explore this possibility, we introduce an attention reward, which quantifies the model's overall visual attention ratio across layers during the reasoning process, as detailed in Appendix A.4. We replace the perception reward with this attention-based reward while keeping all other experimental configurations consistent with the main setup, termed as VAPO$_{attn}$, whereas our original method using perception reward is referred to as VAPO$_{perc}$.

As shown in Table 10, naively maximizing visual attention provides little to no performance gain and even leads to noticeable degradation on certain tasks. We hypothesize that this is due to two key factors: 1) Although visual attention can serve as an indicator of the contribution of visual inputs to the model's decision-making, it does not directly reflect whether the model is effectively utilizing visual features. Blindly encouraging higher attention may disrupt the base model's learned distribution; 2) A higher visual attention ratio is not inherently better. As observed in earlier experiments, its values typically lie between $0$ and $0.3$. Unconstrained maximization of this ratio may lead to instability or training collapse in later stages.

## B.9 COMPUTATIONAL EFFICIENCY AND COST

In this section, we analyze the efficiency and computational cost of our method, and compare it against other policy gradient algorithms in terms of training time and performance gains. We consider two representative reinforcement learning baselines: GRPO and DAPO (Yu et al., 2025). The latter is a widely adopted variant of GRPO that incorporates several advanced techniques, including dynamic sampling and higher clipping ratio. As shown in Table 11, under the same data budget and training epochs, DAPO requires approximately $6$ additional hours of training time compared to GRPO, whereas our method incurs only a marginal increase of around $30$ minutes. More importantly, this modest computational overhead yields a substantial performance gain: our method improves accuracy by $4.37\%$ over GRPO, significantly surpassing the benefits provided by DAPO.

This efficiency is largely attributable to the nature of our perceptual supervision. The key difference of our method from GRPO lies in the anchor scoring process, where the model is asked to evaluate visual claims. Unlike standard rollouts, which require generating full reasoning traces, our method only requires a binary judgment, *i.e.*, yes or no for each claim, essentially a single-token output. Consequently, this process is highly efficient. Moreover, our gradient update procedure remains identical to that of GRPO. These results suggest that our approach introduces minimal additional computational cost while achieving notably greater performance improvements.

Beyond training, In VAPO, we leverage LLMs, typically GPT to generate visual claims for assessing the model's perception capability. Here for completeness, we provide below an estimate of the computational time required to generate these visual claims, based on our usage records. As shown in Table 12, the time is mainly spent on querying GPT and on the subsequent filtering process. The

| Dataset | Total | Querying | Filtering |
|---------|-------|----------|-----------|
| ViRL39K | 38min | 34min | 4min |

Table 12: The time costs for generating visual claims.

querying stage refers to the waiting time after submitting prompts, which also includes re-queries for generated claims that fail to meet the specified requirements. The filtering stage applies a set of rule-based criteria as described in Fig. 10 to remove claims that do not satisfy our standards, for example, those that are excessively long, contain uncertainty markers such as "maybe" or "appears", or violate the required balance between correct and incorrect claims. Besides, we also remove redundant claims that are semantically repetitive with embedding models, and limit the number of claims from various types, *e.g.*, color, relational and OCR, through regular expressions. Any filtered-out examples are resubmitted until the target number of valid claims is reached. Overall, the entire process takes roughly half an hour, which is negligible compared with the training phase and remains highly efficient relative to methods such as DAPO even considering the overall pipeline costs. It is important to note that this claim generation process is a one-time procedure, and no further GPT querying is required for subsequent training runs.

### B.10 LIMITATION ANALYSIS

**Visual Claim Quality**. One factor influencing the effectiveness of VAPO is the quality of the generated visual claims. This includes whether the claims are strongly vision-dependent, clearly verifiable from the image, and correctly labeled. As such, the claim generation model, *i.e.*, GPT-5 in our current setup, may become a limiting factor for overall performance. However, this also suggests that VAPO retains great potential for further improvement: leveraging higher-quality visual claims, either generated by more capable models or annotated by human experts, could potentially enhance the effectiveness and further improve the model's performance ceiling.

**Hyperparameter Sensitivity**. Our method introduces several new hyperparameters including the anchor number $K$, the late-emphasis weight $\beta$, and the perception reward weight $\gamma$, all of which require careful tuning to achieve optimal performance. Moreover, the optimal settings for these hyperparameters may vary across task types: vision-intensive tasks tend to benefit from larger values of $\beta$ and $\gamma$, whereas logic-heavy tasks such as mathematical reasoning often prefer smaller values. Designing an adaptive vision-anchored policy that dynamically adjusts these parameters based on task characteristics remains an important direction for future work.

**Single Image Setting**. In the current work, we focus exclusively on single-image tasks for simplicity. However, VAPO can be readily extended to multi-image or even video-based tasks by generating visual claims that reference multiple frames or views. Exploring the benefits of vision-anchored training in such settings, particularly in multi-image reasoning or temporal video understanding presents an interesting and promising direction for future research.

### B.11 CLAIM AND LABEL QUALITY

Upon obtaining the generated claims, to assess the quality and potential noise of both the visual claims and their corresponding labels, here we conduct a straightforward yet reliable human evaluation as an indicator for their effectiveness prior to training. Specifically, we randomly sample 250 examples from the training set and evaluate each generated claim to determine whether it is visual-dependent, i.e., grounded in the visual cues of the image rather than deducible solely through textual reasoning, and whether it is answerable, i.e., has a clear and unambiguous ground-truth answer. For the labels, we manually verify the accuracy against the corresponding question and visual input. Below, we report the recorded human evaluation statistics for the claims and labels.

As shown in Table 13, the generated claims exhibit high overall quality according to human evaluation, where the noise in the claims occupy only around $2\%$, *i.e.*, claims that are not visual-dependent or not answerable, and the labeling error rate is below $4\%$. In addition, for comprehensiveness, here we also perform model evaluation, in which we sample 2000 examples and feed the generated claims and labels back to GPT and prompt it to act as a judge to assess noise and accuracy as defined

| Human Evaluation | Visual-Dependent (%) | Answerable (%) | Accuracy (%) |
|---|---|---|---|
| Claims | 98.44 | 99.30 | — |
| Labels | — | — | 96.78 |

Table 13: The human audit of the generated claim and label quality.

| Model Evaluation | Visual-Dependent (%) | Answerable (%) | Accuracy (%) |
|---|---|---|---|
| Claims | 97.29 | 99.67 | - |
| Labels | - | - | 95.85 |

Table 14: The model evaluation of the generated claim and label quality.

above. The above results in Table 14 indicate that model and human evaluation of claim quality are closely aligned, suggesting that the generated claims and labels are reliable. As highlighted in our limitation analysis, further reducing the noise and error rates has the potential to yield additional performance gains and further enhance the effectiveness of our approach.

## B.12    METHOD SENSITIVITY TO OTHER GENERATORS

In our main experiments, we employ GPT-5 to generate visual claims due to its strong performance and cost-effectiveness. Here we further examine how using open-source, weaker models to generate visual claims influences VAPO's effectiveness. Specifically, we consider two typical VLMs: LLaVA-1.6-34B (Liu et al., 2023) and Qwen2.5-VL-32B (Bai et al., 2025). For each model, we input the instructions directly as user prompts and apply a filtering mechanism similar to that used with GPT-5 to remove claims that do not meet our requirements. We train VAPO on a subset of 5000 examples randomly sampled from the full training set using the claims generated by these models, and evaluate results on the established benchmarks. As shown in Table 15, despite being open-source models, Qwen2.5-VL achieves performance comparable to those obtained with GPT-5, yielding nearly identical average improvements over GRPO ($\pm 0.2\%$). Similarly, LLaVA-1.6, one of the pioneering VLMs, also attains nearly the full performance achieved with GPT-5 ($\pm 0.5\%$). These results indicate that VAPO does not have to rely on closed-source, state-of-the-art models to be effective; smaller open-source models are capable of delivering almost the same performance gains.

## B.13    VAPO ON OTHER MODELS

Following prior work (Huang et al., 2025; Wang et al., 2025a), we use Qwen2.5-VL as a standard base model to evaluate the effectiveness of our method, given its strong and representative reasoning capabilities. We posit that visual forgetting is primarily driven by two factors: 1) Training data. The existing training corpora (e.g., mathematics or text-intensive tasks) place heavy emphasis on textual patterns, leading models to develop a strong text bias; 2) Training objective. The widespread use of next-token prediction objective encourages models to prioritize autoregressive consistency, causing attention to visual inputs to become progressively diluted as the output sequence grows, as shown in Fig. 3 of the main paper. Therefore, unless these underlying factors are substantially altered, we expect that visual forgetting will likely generalize across different model architectures.

To provide a preliminary validation of this assumption, here we additionally experiment with Qwen3-VL-8B (Yang et al., 2025a), a more recent and more capable reasoning model whose ar-

| Model | MathVerse | LogicVista | Geometry3k | MMMU | MMStar | HallBench | Avg. |
|---|---|---|---|---|---|---|---|
| GRPO | 44.2 | 44.4 | 44.5 | 53.6 | 55.2 | 51.2 | 49.0 |
| LLaVA-1.6 | 48.1 | 46.7 | 47.0 | 56.5 | 58.6 | 55.0 | 52.0 |
| Qwen2.5-VL | 48.5 | 47.0 | 47.4 | 56.7 | 58.9 | 55.2 | 52.3 |
| GPT-5 | 48.9 | 47.3 | 47.7 | 56.7 | 59.1 | 55.5 | 52.5 |

Table 15: The VAPO effectiveness given claims generated with other models.

| Model | MathVerse | LogicVista | Geometry3k | MMMU | MMStar | HallBench | Avg. |
|---|---|---|---|---|---|---|---|
| Qwen3-VL | 58.8 | 56.7 | 59.3 | 68.4 | 67.3 | 59.2 | 61.8 |
| + GRPO | 63.5 | 60.1 | 64.4 | 72.7 | 68.3 | 60.4 | 64.9 |
| + VAPO | **67.6** | **65.8** | **68.7** | **74.2** | **72.5** | **63.9** | **68.8** |

Table 16: The results of VAPO on other base models.

| Model | V* | RealWorldQA | Avg. |
|---|---|---|---|
| Qwen2.5-VL | 71.4 | 64.3 | 67.9 |
| R1-OneVision | 73.5 | 65.7 | 69.6 |
| VLAA-Thinker | 74.8 | 66.3 | 70.6 |
| Vision-R1 | 74.1 | 66.8 | 70.5 |
| VAPO-Thinker | **77.9** | **69.4** | **73.7** |

Table 17: The results of VAPO on more evaluation tasks.

chitecture differ substantially from those of Qwen2.5-VL. We apply VAPO to Qwen3-VL to assess whether our method remains effective under different scenarios. As shown in Table 16, our method continues to provide substantial improvements over GRPO when applied to Qwen3-VL, yielding an average gain of 3.9% (64.9% → 68.8%). This suggests that, beyond Qwen2.5-VL, other reasoning models may also suffer from pronounced visual forgetting, and that our method is broadly applicable and effective across different model architectures.

## B.14 EVALUATION ON MORE TASKS

During training, our visual claims are designed to span multiple dimensions of perceptual capability, including fine-grained visual details, *e.g.*, "the whiteboard says No Parking", and spatial reasoning, *e.g.*, "the bus is to the left of the motorcycle". Consequently, our perception training enhances performance on tasks requiring these capabilities. For completeness, here we further evaluate our model on two popular benchmarks, V* (Wu & Xie, 2024) and RealWorldQA, which target fine-grained perception and spatial reasoning, respectively. As shown in Table 17, our model achieves an average improvement of 3.1% (70.6% → 73.7%) over the previous best results on both fine-grained and spatial reasoning benchmarks. This empirically demonstrates that VAPO generalizes well to a broad range of perception-intensive tasks.

## B.15 VAPO WITH SELF-SUPERVISED ANCHORS

The core idea of VAPO is to utilize anchors to assess model's perception capability during reasoning process and use curated rewards to encourage stronger reliance on visual inputs. The specific form of the anchor (i.e., the mechanism used to measure perception), is not essentially limited to binary textual claims. We choose textual claims mainly because 1) efficiency: As discussed in the efficiency analysis, current pipeline introduces only minimal training overhead, offering an excellent cost-performance trade-off; 2) simplicity, unlike reconstruction-based methods which may require additional learnable decoders or more complex architectures, textual claims allow us to probe perception capability with only a single-token output; 3) effectiveness: Our experiments also demonstrate that textual claims significantly improve model accuracy, surpassing previous strong baselines.

When designing the visual anchors, we also considered reconstruction-based anchors, where the model would be required to reconstruct visual features or undergo linear probing to assess its per-

| Model | MathVerse | LogicVista | Geometry3k | MMMU | MMStar | HallBench | Avg. |
|---|---|---|---|---|---|---|---|
| Base Model | 40.7 | 42.6 | 38.5 | 52.7 | 54.9 | 50.0 | 46.6 |
| GRPO | 44.2 | 44.4 | 44.5 | 53.6 | 55.2 | 51.2 | 49.0 |
| VAPO (Self-Supervised) | 48.1 | 46.5 | 48.3 | 55.9 | 57.9 | 55.1 | 51.9 |
| VAPO (GPT) | 48.9 | 47.3 | 47.7 | 56.7 | 59.1 | 55.5 | 52.5 |

Table 18: The results of VAPO with self-supervised anchors.

| Cut Position (%) | 20 | 40 | 60 | 80 | Agg. |
|---|---|---|---|---|---|
| Recoverable Ratio (%) | 4.5 | 5.1 | 8.8 | 7.6 | 15.4 |

Table 19: The single-cut recoverable ratio by early decision.

ceptual capability. However, as mentioned above, these approaches introduce additional learnable parameters, thereby increasing computational and training complexity.

Another promising self-supervised direction is to adopt a bootstrapping approach, in which the model generates its own visual claims and subsequently uses them as training signals, thereby eliminating the need for external models. Here we explore whether this more efficient training strategy for VAPO remains effective. Specifically, instead of relying on GPT to produce visual claims, we allow the model being trained to generate its own visual claims during the rollout step using the provided instructions, and then use these self-generated claims to assess its perceptual capability and update the model accordingly. This procedure is self-supervised and iterative: the model's generated claims help improve its visual capability, which in turn further refines the quality of subsequent claims. We use Qwen2.5-VL-7B as the base model and train on a 5000 example subset of the full training set. As shown in Table 18, this self-supervised approach retains the vast majority of the effectiveness observed in the default (GPT) setting, with only a modest average difference of 0.7% ($46.5\% \rightarrow 47.3\%$). This indicates that, even without relying on any external or stronger models, VAPO can still achieve highly promising performance.

### B.16 SINGLE-CUT RECOVERABLE RATIO

Our current definition of the recoverable ratio uses a relatively loose metric intended to estimate the proportion of error cases influenced by the reasoning process, serving as an oracle-style indicator of the potential for our proposed method. Here, we additionally report recoverable ratios at specific single-cut positions. In particular, we provide the recoverable ratios measured at four relative reasoning depths (20%, 40%, 60%, and 80% of the reasoning length) as well as their aggregated result. As shown in Table 19, even when considering only a single reasoning position, the model's recoverable ratio remains substantial, reaching nearly 10% at the 60% and 80% positions. This indicates considerable headroom for improving the accuracy of current models and helps explain why excessively long reasoning can degrade performance, as observed in Fig. 2. Moreover, we observe that later positions exhibit higher recoverable ratios, suggesting that the latter stages of reasoning have a particularly strong impact on the model's reliance on visual information, consistent with the visual forgetting phenomenon shown in Fig. 3.

### B.17 DATA DEDUPLICATION CHECK

In our training setup, we follow prior work (Wang et al., 2025a;d) by training on ViRL39K and evaluating on both mathematical benchmarks (e.g., MathVista) and general-purpose benchmarks (e.g., MMStar). Here we perform data deduplication checks between the training set and the evaluation benchmarks. Specifically, we compare image hash values across datasets and compute the number of overlapping instances. As shown in Table 20, both the mathematical benchmarks (6 in total) and the general-purpose benchmarks (4 in total) exhibit 0% overlap with the training dataset based on image-hash comparison. This indicates that no duplicate samples exist between the training and evaluation sets, and thus there is no risk of data contamination.

| Deduplication Check | Mathematical | General | Total |
|---|---|---|---|
| Overlap Ratio (%) | 0 | 0 | 0 |

Table 20: The overlap ratio between training and evaluation set.

| Training Steps | 0 | 25 | 50 | 75 | 100 |
|---|---|---|---|---|---|
| Activation Rate (Easy) | 0.53 | 0.65 | 0.72 | 0.76 | 0.79 |
| Activation Rate (Hard) | 0.21 | 0.44 | 0.56 | 0.61 | 0.64 |

Table 21: The accuracy gate activation gate across easy and hard examples.

| Method | MathVerse | LogicVista | Geometry3k | MMMU | MMStar | HallBench | Avg. |
|---|---|---|---|---|---|---|---|
| GRPO | 44.2 | 44.4 | 44.5 | 53.6 | 55.2 | 51.2 | 49.0 |
| VAPO (strict gate) | **48.9** | 47.3 | **47.7** | **56.7** | 59.1 | **55.5** | **52.5** |
| VAPO (relaxed gate) | 48.5 | **47.4** | 47.2 | 56.3 | **59.4** | 55.0 | 52.3 |

Table 22: The results of the curriculum gating strategy.

## B.18 EFFECT OF ACCURACY GATE

In training, we incorporate an accuracy gate to prevent reward hacking, where the model might pursue perception rewards at the expense of task accuracy by producing trivially short reasoning. Although the accuracy gate is triggered more frequently on easy problems, resulting in a larger portion of trajectories receiving perception rewards, these rewards are ultimately normalized when converted into advantages. This normalization step ensures that the generally higher rewards obtained from easy examples do not disproportionately influence the final advantage. In contrast, for hard problems, normalization scales up the advantages of correct trajectories, encouraging the model to pay additional attention to these challenging cases. Therefore, the accuracy gate does not bias training toward simpler examples.

To further analyze how the accuracy gate influences the training dynamics of easy and hard problems, here we report the accuracy gate activation rates over the course of training for both categories. We classify problems as easy or hard based on the model's initial accuracy gate activation rate ($> 0.4$ for easy and $< 0.4$ for hard) and track how these activation rates evolve at different training steps. We conduct training on a $5000$ example subset of the full training set. As shown in Table 21, the activation rate increases substantially for both easy and hard questions, and the gap between the two progressively narrows. This further indicates that the model does not over or under-optimize either category during training, suggesting balanced learning dynamics across problem difficulties.

Moreover, to investigate whether a relaxed accuracy gate can further improve performance, we experiment with a curriculum gating strategy. Specifically, unlike our default setting, which applies a strict binary gate, we gradually loosen this constraint over the course of training. That is, we progressively assign an increasing proportion of the perception reward to incorrect trajectories and eventually remove the accuracy gate entirely by granting full perception rewards. As before, we conduct this study using a $5000$ example subset of the full training set. As shown in Table 22, using a relaxed accuracy gate does not yield noticeable performance improvements. This aligns with our earlier analysis that the default accuracy gate does not under-optimize hard examples, and that relaxing the gate introduces a greater risk of reward hacking.

## B.19 EFFECT OF ANCHOR PLACEMENT STRATEGY

Our current anchor placement strategy resembles the early decision scheme, where anchors are inserted at semantic boundaries (e.g., commas, periods, line breaks) randomly throughout the entire reasoning trajectory. In designing this component, we also considered several alternative placement strategies: 1) uniform, where anchors are inserted at fixed intervals rather than randomly; 2) backloaded, where anchors are inserted only in the latter $60\%$ of the reasoning process, acting as a targeted mechanism to address severe visual forgetting that tends to occur toward the end of long trajectories; 3) front-loaded, where anchors are inserted only within the first $60\%$ of the trajectory, serving as a baseline comparison. We ensure that all three variants insert the same number of anchors ($K = 20$). We train each variant on a $5000$ example subset of the full training set, and we report the results of these placement strategies.

| Position | MathVerse | LogicVista | Geometry3k | MMMU | MMStar | HallBench | Avg. |
|---|---|---|---|---|---|---|---|
| GRPO | 44.2 | 44.4 | 44.5 | 53.6 | 55.2 | 51.2 | 49.0 |
| Uniform | 48.4 | 47.1 | 47.5 | 56.1 | 58.4 | 55.0 | 52.1 |
| Front-loaded | 48.1 | 46.7 | 47.0 | 55.9 | 58.5 | 54.4 | 51.8 |
| Back-loaded | 48.7 | 46.9 | 47.7 | 56.2 | 59.0 | **55.8** | 52.4 |
| Random | **48.9** | **47.3** | **47.7** | **56.7** | **59.1** | 55.5 | **52.5** |

Table 23: The effect of varying anchor placement strategies.

| Method | MathVerse | LogicVista | Geometry3k | MMMU | MMStar | HallBench | Avg. |
|---|---|---|---|---|---|---|---|
| Base Model | 40.7 | 42.6 | 38.5 | 52.7 | 54.9 | 50.0 | 46.6 |
| MM-Eureka | 50.3 | 48.5 | 50.2 | 55.7 | 60.4 | 54.8 | 53.3 |
| PAPO | 50.4 | 47.3 | 49.7 | **60.8** | 61.8 | 55.6 | 54.2 |
| VL-Rethinker | **54.2** | 47.2 | 49.6 | 57.9 | 61.8 | 55.4 | 54.4 |
| VAPO | 53.3 | **50.9** | **51.3** | 60.2 | **63.0** | **57.4** | **56.0** |

Table 24: The comparison between VAPO with more baselines.

As shown in Table 23, the simple naive (random) strategy performs better than other placement variants. We attribute this to several factors: 1) uniform insertion, which places anchors at fixed intervals, is more susceptible to reward hacking, as the model may learn to exhibit high visual dependence only at predetermined positions; 2) front-loaded insertion performs noticeably worse than both random and back-loaded insertion, indicating that later positions in the reasoning trajectory are more critical for improving perceptual grounding; 3) In addition, although back-loaded insertion is slightly weaker than random insertion on average, it achieves comparable or even superior performance on MMStar and HallusionBench. This suggests that vision-heavy benchmarks may benefit from a larger number of anchors or greater weighting at later reasoning steps, which aligns with our late-emphasis design.

## B.20    COMPARISON WITH MORE BASELINES

Here, for completeness, we compare our approach with more baselines, including PAPO (Wang et al., 2025d), MM-Eureka (Meng et al., 2025) and VL-Rethinker (Wang et al., 2025a). We use the publicly released 7B checkpoints from their repositories. For PAPO, we reproduce the results using greedy decoding, as their paper reports performance using the avg@8 accuracy metric, which is not directly comparable to the metrics adopted in prior work (Wang et al., 2025a; Huang et al., 2025). As shown in Table 24, VAPO achieves an average improvement of $1.6\%$ over the previous best results ($54.4\% \rightarrow 56.0\%$), further demonstrating the effectiveness of our approach.

## B.21    EFFECT OF VAPO ON LONG-THOUGHT EXAMPLES

Our ablation study in Fig. 7 shows that increasing $K$, $i.e.$, using more anchors, generally yields positive gains across examples. To further examine whether longer-thought examples benefit disproportionately from larger $K$, we partition benchmark examples into two groups based on the output length of our VAPO-Thinker-7B model: short-thought ($< 300$ tokens) and long-thought ($> 300$ tokens). We then evaluate models trained with different values of $K$ and analyze how performance trends vary across these two categories. As shown in Table 25, increasing $K$ leads to stable performance improvements for both short- and long-thought examples, and the magnitude of these gains

| Length | 1 | 5 | 10 | 15 | 20 |
|---|---|---|---|---|---|
| Short | 53.7 | 56.0 | 57.5 | 58.4 | 58.5 |
| Long | 47.3 | 49.4 | 50.9 | 51.5 | 51.8 |
| Total | 51.5 | 53.8 | 55.1 | 55.8 | 55.9 |

Table 25: The results of VAPO on short and long-thought examples.

| Models | MathVerse | LogicVista | Geometry3k | MMMU | MMStar | HallBench | Avg. |
|---|---|---|---|---|---|---|---|
| R1-OneVision | 299 | 308 | 287 | 263 | 232 | 201 | 265 |
| Vision-R1 | 337 | 293 | 341 | 283 | 239 | 251 | 291 |
| VAPO-Thinker | 315 | 287 | 334 | 275 | 254 | 238 | 284 |

Table 26: The average reasoning length of VAPO.

| Method | BLINK | MuirBench | Avg. |
|---|---|---|---|
| Base model | 55.3 | 58.1 | 56.7 |
| GRPO | 57.0 | 60.4 | 58.7 |
| VAPO | 60.1 | 62.2 | 61.2 |

Table 27: The results of VAPO on multi-image tasks.

remains largely consistent across the two groups. This trend is also aligned with the ablation results reported in Fig. 7.

### B.22 REASONING LENGTH DISTRIBUTION

The goal of VAPO is to enhance the effectiveness of reasoning by improving the model's perceptual capability, rather than to alter the length of its reasoning traces. Accordingly, our pipeline does not incorporate any explicit length regularization. Here we report the average reasoning length of our models compared with other baselines across the established benchmarks. As shown in Table 26, the average output length of our model does not differ noticeably from that of other baselines. This rules out the possibility that our performance gains stem merely from longer reasoning traces, and instead indicates that VAPO improves the effectiveness of the reasoning process itself.

### B.23 VAPO ON MULTI-IMAGE TASK

Our current visual claims are primarily designed for single-image inputs, which is sufficient to validate the core motivation of VAPO, namely improving the model's reliance on visual information and mitigating visual forgetting. Here we further conduct a small-scale pilot study on multi-image tasks. Specifically, we train VAPO exclusively on the multi-image subset of ViRL39K, which contains approximately 2,500 examples. For claim generation, we consider two types of claims. The first type is single-image claims, which mirror our original setup and are generated with respect to only one image in the multi-image set. The second type is cross-image claims, which capture relationships or distinctions across images. For the former, we provide GPT with only one image at a time. For the latter, we present all images simultaneously and extend the original instructions with an additional requirement that the generated claim must reference information across images (e.g., "the two images depict the same building"). We ensure that each training example contains a balanced number of claims from both categories. The training procedure remains identical to our main pipeline, and GRPO is used as the baseline for comparison. We evaluate the trained models on two representative multi-image benchmarks, including BLINK (Fu et al., 2024) and MuirBench (Wang et al., 2024a).

As shown in Table 27, despite being trained on a limited amount of data, our method still achieves an average improvement of 2.5% over GRPO on multi-image tasks (58.7% → 61.2%). This demonstrates that our approach is not restricted to the single-image setting and can generalize to multiple images as well. We leave the evaluation of video-based tasks as promising future work.

## C. BROADER SOCIETAL IMPACTS

Our work carries several positive societal implications. Our findings reveal that existing VLMs often produce reasoning that is not visually grounded, which can degrade performance and reliability. Such ungrounded reasoning may introduce misleading or hallucinated content, thereby posing risks to model safety and trustworthiness. Moreover, since these reasoning traces often rely heavily on statistical patterns learned from training data, they may inadvertently expose sensitive information,

raising concerns about data privacy and potential leakage. In contrast, the proposed `VAPO` framework steers the model's reasoning process to remain anchored in visual evidence. This grounding mechanism helps ensure that model outputs are more faithful to the input image and less likely to include irrelevant or speculative content, thus promoting both factual accuracy and privacy preservation. At present, we have not identified negative societal impacts associated with this work. However, due to external factors such as the availability of datasets and baseline implementations, further assessment may be necessary in the future.

## D. SUPPLEMENTARY RESULTS

### D.1 FULL NUMERICAL MAIN RESULTS

Due to space limitations in the main text, some baseline results are omitted. Here, we provide the full numerical results of our method and all baselines across the ten established benchmarks in Table 28, corresponding to Table 1 and Table 2.

### D.2 NUMERICAL RESULTS OF $K$

Here we report the numerical results of the ablation study on the anchor number $K$ in Table 29, corresponding to the visualization in Fig. 7 (A).

### D.3 NUMERICAL RESULTS OF $\beta$

Similarly, we present the numerical results of the ablation study on the impact of $\beta$ in Table 30, corresponding to Fig. 7 (B).

### D.4 FULL RESULTS OF AUGMENTED BASELINES

Here we provide the full numerical results of our method compared with baselines augmented with inference-level remedies, *i.e.*, visual replay and focus prompt, in Table 31, corresponding to Table 3 in the main paper.

### D.5 FULL RESULTS OF 3B PARAMETER SCALE

Here for completeness, we report the evaluation results of our proposed method and baseline models at the 3B scale. To ensure fairness, we select baselines with comparable model sizes and exclude those without publicly available 3B-scale checkpoints. The results are presented in Table 32.

| Models | MathVerse | MathVista | MathVision | LogicVista | WeMath | Geo3k | MMMU | MMStar | HallBench | MMVet | Avg. |
|---|---|---|---|---|---|---|---|---|---|---|---|
| *Close-source Models* | | | | | | | | | | | |
| GPT-5-Thinking | 81.2 | 81.9 | 72.0 | 70.0 | 71.1 | 79.9 | 81.8 | 75.7 | 65.2 | 77.6 | 75.6 |
| Gemini-2.5-Pro | 76.9 | 80.9 | 69.1 | 73.8 | 78.0 | 77.2 | 74.7 | 73.6 | 64.1 | 83.3 | 75.2 |
| *Open-source Models* | | | | | | | | | | | |
| Qwen2.5-VL-7B | 40.7 | 62.3 | 23.2 | 42.6 | 33.1 | 38.5 | 52.7 | 54.9 | 50.0 | 64.8 | 46.3 |
| InternVL2.5-8B | 34.5 | 68.2 | 25.6 | 38.3 | 38.6 | 44.8 | 56.2 | 63.2 | 49.0 | 62.8 | 48.1 |
| R1-OneVision-7B | 46.4 | 64.1 | 29.9 | 45.6 | **44.6** | 46.1 | 54.3 | 54.1 | 52.5 | 65.2 | 50.3 |
| VLAA-Thinker-7B | 48.2 | 68.0 | 26.4 | 48.5 | 41.5 | 50.6 | 59.1 | 49.7 | 54.7 | 70.0 | 51.7 |
| Vision-R1-7B | 52.4 | 73.5 | 28.2 | 49.7 | 41.6 | 49.0 | 57.6 | 61.4 | 49.5 | 71.1 | 53.4 |
| *Our Models* | | | | | | | | | | | |
| VAPO-Thinker-3B | 35.8 | 67.1 | 23.9 | 39.7 | 35.4 | 44.2 | 55.6 | 59.4 | 49.5 | 64.6 | 47.5 |
| VAPO-Thinker-7B | **53.3** | **75.6** | **31.9** | **50.9** | 43.6 | **51.3** | **60.2** | **63.0** | **57.4** | 71.9 | **55.9** |

Table 28: The full numerical results of main experiments, corresponding to Table 1 and Table 2.

| $K$ | MathVerse | MathVista | MathVision | LogicVista | WeMath | Geo3k | MMMU | MMStar | HallBench | MMVet | Avg. |
|----|-----------|-----------|------------|------------|--------|-------|------|--------|-----------|-------|------|
| 0  | 48.2 | 70.6 | 26.1 | 45.5 | 39.1 | 47.3 | 56.6 | 58.9 | 53.2 | 69.9 | 51.5 |
| 5  | 50.9 | 73.1 | 28.4 | 47.8 | 41.2 | 50.5 | 58.4 | 61.3 | 55.6 | 70.4 | 53.8 |
| 10 | 52.8 | 74.8 | 30.7 | 49.5 | 42.5 | 51.2 | 59.9 | 62.2 | 56.7 | 71.2 | 55.1 |
| 15 | 53.2 | **75.8** | 31.5 | 50.4 | **43.9** | 51.0 | **60.4** | 62.7 | 57.0 | 71.6 | 55.8 |
| 20 | **53.3** | 75.6 | **31.9** | **50.9** | 43.6 | **51.3** | 60.2 | **63.0** | **57.4** | **71.9** | **55.9** |

Table 29: The numerical results of ablation study of $K$, corresponding to Fig. 7 (A).

| $\beta$ | MathVerse | MathVista | MathVision | LogicVista | WeMath | Geo3k | MMMU | MMStar | HallBench | MMVet | Avg |
|---|---|---|---|---|---|---|---|---|---|---|---|
| 0.0 | 51.7 | 73.8 | 28.4 | 48.6 | 42.0 | 49.9 | 58.9 | 60.2 | 53.8 | 70.3 | 53.8 |
| 0.5 | 52.7 | 74.8 | 30.2 | 49.6 | 42.8 | 50.7 | 59.5 | 62.0 | 55.4 | 71.3 | 54.9 |
| 1.0 | 53.1 | **75.7** | 31.5 | **51.3** | 43.4 | 51.4 | 60.0 | 62.5 | 56.8 | **72.0** | 55.8 |
| 1.5 | **53.3** | 75.6 | **31.9** | 50.9 | **43.6** | 51.3 | **60.2** | 63.0 | 57.4 | 71.9 | **55.9** |
| 2.0 | 53.0 | 75.5 | 31.1 | 50.5 | 43.1 | **51.5** | 60.0 | **63.3** | **56.6** | 71.6 | 55.6 |
| 2.5 | 51.8 | 74.8 | 30.4 | 50.0 | 42.6 | 50.5 | 59.8 | 62.2 | 55.3 | 70.2 | 54.7 |

Table 30: The numerical results of ablation study of $\beta$, corresponding to Fig. 7 (B).

| Models | MathVerse | MathVista | MathVision | LogicVista | WeMath | Geo3k | MMMU | MMStar | HallBench | MMVet | Avg |
|---|---|---|---|---|---|---|---|---|---|---|---|
| VLAA-Thinker-7B | 48.2 | 68.0 | 26.4 | 48.5 | 41.5 | 50.6 | 59.1 | 49.7 | 54.7 | 70.0 | 51.7 |
| + Focus Prompt | 48.8 | 68.4 | 27.3 | 49.2 | 41.8 | 50.7 | 59.7 | 51.1 | 55.2 | 70.7 | 52.3 |
| + Visual Replay | 49.8 | 69.8 | 27.9 | 49.9 | 42.3 | 51.1 | 60.5 | 52.9 | 56.2 | 71.7 | 53.2 |
| Vision-R1-7B | 52.4 | 73.5 | 28.2 | 49.7 | 41.6 | 49.0 | 57.6 | 61.4 | 49.5 | 71.1 | 53.4 |
| + Focus Prompt | 52.8 | 73.8 | 29.3 | 50.2 | 42.1 | 49.7 | 58.7 | 61.8 | 50.5 | 71.4 | 54.0 |
| + Visual Replay | **53.4** | 75.2 | 29.9 | 50.7 | 42.5 | 50.5 | 59.4 | 62.1 | 51.8 | 71.9 | 54.7 |
| VAPO-Thinker-7B | 53.3 | **75.6** | **31.9** | **50.9** | **43.6** | **51.3** | **60.2** | **63.0** | **57.4** | **71.9** | **55.9** |

Table 31: The full results of baselines augmented with test-time remedies, corresponding to Table 3.

| Models | MathVerse | MathVista | MathVision | LogicVista | WeMath | Geo3k | MMMU | MMStar | HallBench | MMVet | Avg. |
|---|---|---|---|---|---|---|---|---|---|---|---|
| Qwen2.5-VL-3B | 30.5 | 60.3 | 19.8 | 37.6 | 20.4 | 31.7 | 48.6 | 51.5 | 44.0 | 58.4 | 40.3 |
| InternVL2.5-2B | 22.3 | 51.1 | 14.0 | 27.3 | 8.0 | 36.1 | 43.2 | 54.3 | 42.3 | 62.6 | 36.1 |
| R1-OneVision-3B | 35.5 | — | 23.7 | — | — | — | — | — | — | — | — |
| VLAA-Thinker-3B | 36.4 | 61.0 | 24.4 | 38.5 | 33.8 | 42.7 | 50.5 | 56.2 | 45.1 | 63.9 | 45.3 |
| VAPO-Thinker-3B | **38.2** | **64.5** | **27.3** | **41.4** | **35.3** | **44.2** | **51.6** | **59.4** | **49.5** | **64.6** | **47.6** |

Table 32: The results of our 3B model and baselines of comparable scale. "—" indicates unavailable results due to unreleased checkpoints. Vision-R1 is excluded as it does not consider 3B scale.

