# OpenReview forum: "More Thought, Less Accuracy? On the Dual Nature of Reasoning in Vision-Language Models"
_ICLR.cc/2026/Conference — ICLR 2026 Poster_

### Official Review · Reviewer_j11z · 2025-10-29

**Soundness:** 3
**Presentation:** 4
**Contribution:** 4
**Rating:** 6
**Confidence:** 4

**Summary:**

The paper investigates the dual nature of multimodal reasoning in vision-language models, showing that while early reasoning often improves performance, prolonged reasoning can weaken perceptual grounding and reduce accuracy on visually simple questions. Using an “early decision” protocol that forces answers at intermediate points, the authors find that accuracy first increases and then declines as reasoning continues. Error analysis shows perception errors dominate and many could be avoided by stopping earlier. This phenomenon is attributed to visual forgetting, where attention to visual tokens decays over time. Two inference-time fixes—visual replay and focus prompts—temporarily restore visual attention and improve accuracy. To address the issue during training, the authors propose VAPO (Vision-Anchored Policy Optimization), which inserts visual claim anchors along the reasoning process and introduces a perception reward that encourages correct yes/no judgments of image-grounded claims. With late-emphasis weighting and an accuracy-gated reward, VAPO-Thinker-7B achieves state-of-the-art results on multiple benchmarks, particularly those requiring strong visual grounding, while maintaining efficiency. Ablations confirm the effectiveness of more anchors, later-anchor emphasis, stable training dynamics, and superiority over attention-only rewards and inference-time fixes.

**Strengths:**

- Original analysis of multimodal reasoning dynamics: the early-decision protocol provides a fine-grained view of how different reasoning segments contribute to final accuracy, revealing a non-monotonic curve where longer reasoning can hurt (Sec. 3.1, Fig. 2A) and that the harm is largely perceptual (Fig. 2B–C).
- Clear causal story with measurements and interventions: the visual-attention decay analysis (Fig. 3A) and simple test-time remedies (visual replay, focus prompt) that immediately spike attention and improve accuracy (Fig. 3B) support the visual forgetting hypothesis (Sec. 3.2).
- Methodological novelty and simplicity: VAPO’s claim-anchored perception reward operationalizes “visual grounding” during RL without architectural changes. The late-emphasis weighting targets the observed late-stage forgetting (Sec. 4.2, Eq. 3–4).
- Strong empirical results and breadth: consistent gains across 10 benchmarks with notable improvements on vision-intensive datasets (MMStar, HallusionBench) and competitive math performance (Tables 1–2, 11). Thorough ablations on anchor number K, weighting β, reward weight γ, KL coefficient, and baselines with test-time remedies (Fig. 7; Tables 3, 4, 7–9, 14).
- Efficiency: small training-time overhead over GRPO and larger accuracy gains than a stronger RL baseline (DAPO) (Table 10).
- Analysis depth: learning curves (Fig. 6), attention trends versus baseline (Fig. 5A), accuracy versus reasoning progress (Fig. 5B), and anchor-score diagnostics (Fig. 13) collectively strengthen the argument.

**Weaknesses:**

- Dependence on a closed-source claim generator (GPT-5) and limited auditing of claim correctness and vision-dependency. While Appendix A.6 provides prompts and Table 6 explores variants, there is no quantitative estimate of claim label noise or a human audit; method sensitivity to generator quality is acknowledged but not systematically evaluated.
- The “recoverable by early decision” metric aggregates across multiple potential cut points (Fig. 2 caption) and may overestimate recoverability. A stricter, single-cut metric or randomized stopping analysis would better calibrate the effect size.
- Potential overlap and contamination risks are not fully ruled out: ViRL39K composition (Sec. 5) merges several sources; explicit deduplication against evaluation sets is not reported. This is especially relevant for MMStar and MathVista where many public pipelines exist.
- The accuracy-gated perception reward can bias learning toward easier instances where final answers are already correct, potentially under-incentivizing perception on hard items. An ablation that relaxes the gate or uses shaped rewards conditioned on partial correctness would clarify trade-offs.
- Anchor placement strategy is mostly random with late weighting; more analysis of position selection (e.g., semantic segment boundaries versus uniform) would strengthen claims about targeting late-stage forgetting beyond the β ablation (Fig. 7B).
- Comparison to closely related RL formulations emphasizing perception (e.g., perception-aware policy optimization [Wang et al., 2025c]) could be deepened to delineate conceptual/implementation differences and when claim-anchoring is preferable.
- The attention metric is an indirect proxy; while useful, further triangulation (e.g., causal masking of image tokens at anchor points, feature attribution) would reduce the risk of over-interpreting attention as reliance.
- Some related works about RL+Reasoning in VLM are missed.
[1] Wang, Jiaqi, et al. "Think or Not? Selective Reasoning via Reinforcement Learning for Vision-Language Models." arXiv preprint arXiv:2505.16854 (2025).

**Questions:**

- Claim quality and robustness:
  - What fraction of GPT-5 claims were mislabeled upon manual audit? Can you report inter-annotator agreement or a small-scale human validation of claim correctness and vision-dependency (Sec. A.6; Table 6)?
  - How sensitive is VAPO to weaker generators? Please provide results with an open-source vision-capable LLM to generate claims and quantify the drop from GPT-5.
  - Do you filter claims for redundancy or triviality (e.g., color naming in synthetic diagrams)? Any diversity constraints?
- Reward design:
  - How often does the accuracy gate mask perception reward during training? A histogram versus steps would reveal whether hard items are under-optimized.
  - Could curriculum gating (e.g., gradually reducing the accuracy condition) further improve performance on harder instances?
  - Have you tried per-claim calibrated rewards (e.g., confidence-weighted yes/no) or multi-class verification to reduce sparsity?
- Anchor placement:
  - Beyond late-emphasis β, did you compare placement by semantic boundaries (sentence/step ends) versus uniform random? Any gains on MMStar/HallusionBench?
  - How does K interact with sequence length at test time? Do longer-thought samples benefit disproportionately from more anchors?
- Evaluation rigor:
  - Please provide explicit data deduplication checks between ViRL39K and each benchmark, including image hash and near-duplicate detection.
  - The “recoverable” metric (Fig. 2B–C) aggregates over multiple early stops. Can you report accuracy when stopping at a single, pre-specified percentile for all samples (e.g., 40% of tokens), to avoid optimistic selection?
  - Reasoning length distribution: does VAPO change average reasoning length versus baselines? If so, how does that correlate with attention and accuracy?
- Comparisons and scope:
  - Clarify distinctions from perception-aware policy optimization (Wang et al., 2025c): is the novelty primarily the claim-anchored verification and late-weighting? Could their reward be used as an auxiliary alongside anchors?
  - Any results on multi-image or video reasoning using multi-frame claims (future Sec. B.10 suggests feasibility)? A small-scale pilot would be informative.

---

> ### Author Response · Authors · 2025-11-21
> **Response to Reviewer j11z by Authors (1/8)**
>
> We sincerely appreciate your valuable comments and recognition of our work. Below are our responses to your concerns.
>
> >**W1: The human audit and method sensitivity of claim/label quality**
>
> **For quantitative estimation of the generated claims and labels**, we conducted a human audit upon claim generation as an indicator of their quality and suitability prior to training. Specifically, for each visual claim, we assess whether it is: 1) *visual-dependent*: grounded in the visual cues of the image rather than derivable solely through textual reasoning; and 2) *answerable*: has a clear and unambiguous ground-truth answer. For the labels, we manually verified their correctness by cross-checking them against the corresponding question and visual input. We randomly sampled 250 examples from the dataset, and below we report the recorded human evaluation statistics for both the claims and the labels.
>
> | Human Evaluation | Visual-Dependent (%) | Answerable (%) | Accuracy (%) |
> |:----------------:|:-------------------:|:-------------:|:-----------:|
> |      Claims      |        98.44        |     99.30     |     -     |
> |      Labels      |         -         |      -      |    96.78    |
>
> As shown in the table above, the generated claims exhibit high overall quality according to human evaluation, with noise (i.e., claims that are not visual-dependent or not answerable) accounting for less than 2%. For label correctness, the GPT-generated labels achieve an accuracy of approximately 97%, corresponding to an error rate of under 4%. For soundness, here we also conduct a model evaluation: we sample 2000 examples, feed the generated claims and labels back into the GPT model, and prompt it to act as a judge to assess noise and accuracy using the same criteria described above.
>
> | Model Evaluation | Visual-Dependent (%) | Answerable (%) | Accuracy (%) |
> |:----------------:|:-------------------:|:-------------:|:-----------:|
> |      Claims      |        97.29        |     99.67     |     -     |
> |      Labels      |         -         |      -      |    95.85    |
>
> The results above show that model-based and human evaluations of claim quality are highly consistent, indicating that the generated claims and labels are reliable. As noted in our limitation analysis in Appendix B.10, further reducing the remaining noise and error rates may unlock additional performance gains and further improve the overall effectiveness of our approach. We have included this experiment in Appendix B.11 of the revised version (highlighted in blue).
>
> **For method sensitivity to claim/label quality**, we employ GPT-5 to generate visual claims in our main experiments due to its strong performance and cost-effectiveness. To assess the sensitivity of VAPO to the quality of claims and labels, we further investigate how using open-source, weaker models affects its effectiveness. Specifically, we consider two representative VLMs: LLaVA-1.6-34B and Qwen2.5-VL-32B. For each model, we provide the same instruction prompts used with GPT-5 (as shown in Fig. 10 in the Appendix). Due to time constraints, we train VAPO on a subset of 5000 examples randomly sampled from the full training set using the claims generated by these models, and we evaluate the resulting models on the established benchmarks.
>
> |    Model   | MathVerse | LogicVista | Geometry3k | MMMU | MMStar | HallBench | Avg. |
> |:----------:|:---------:|:----------:|:----------:|:----:|:------:|:---------:|:----:|
> |    GRPO    |    44.2   |    44.4    |    44.5    | 53.6 |  55.2  |    51.2   | 49.0 |
> |  LLaVA-1.6 |    48.1   |    46.7    |    47.0    | 56.5 |  58.6  |    55.0   | 52.0 |
> | Qwen2.5-VL |    48.5   |    47.0    |    47.4    | 56.7 |  58.9  |    55.2   | 52.3 |
> |    GPT-5   |    48.9   |    47.3    |    47.7    | 56.7 |  59.1  |    55.5   | 52.5 |
>
> As shown in the table, despite being open-source models, Qwen2.5-VL achieves performance comparable to that obtained with GPT-5, yielding nearly identical average improvements over GRPO (±0.2%). Likewise, LLaVA-1.6, one of the pioneering VLMs, attains almost the full performance achieved with GPT-5 (±0.5%). These findings indicate that VAPO does not require closed-source, state-of-the-art models to be effective; smaller open-source models can deliver nearly the same performance gains. This also demonstrates the robustness of VAPO to variations in claim and label quality. We have included this experiment in Appendix B.12 of the revised version (highlighted in blue).

---

> ### Author Response · Authors · 2025-11-21
> **Response to Reviewer j11z by Authors (2/8)**
>
> >**W2: The single-cut recoverable ratio by early decision**
>
> Our current definition of the recoverable ratio uses a relatively loose metric intended to estimate the proportion of error cases influenced by the reasoning process, serving as an oracle-style indicator of the potential for our proposed method. As suggested by the reviewer, we additionally report recoverable ratios at specific single-cut positions. In particular, we provide the recoverable ratios measured at four relative reasoning depths (20%, 40%, 60%, and 80% of the reasoning length) as well as their aggregated result.
>
> |    Cut Position (%)   |  20 |  40 |  60 |  80 | Agg. |
> |:---------------------:|:---:|:---:|:---:|:---:|:----:|
> | Recoverable Ratio (%) | 4.5 | 5.1 | 8.8 | 7.6 | 15.4 |
>
> As shown in the results, even when considering only a single reasoning position, the model’s recoverable ratio remains substantial, reaching nearly 10% at the 60% and 80% positions. This indicates considerable headroom for improving the accuracy of current models and helps explain why excessively long reasoning can degrade performance, as observed in Fig. 2(A). Moreover, we observe that later positions exhibit higher recoverable ratios, suggesting that the latter stages of reasoning have a particularly strong impact on the model’s reliance on visual information, consistent with the visual forgetting phenomenon shown in Fig. 3. In the revised paper, we have included this experiment in Appendix B.16 (highlighted in blue).
>
> >**W3: The potential overlap between training and evaluation set**
>
> In our training setup, we follow prior work [A, B] by training on ViRL39K and evaluating on both mathematical benchmarks (e.g., MathVista) and general-purpose benchmarks (e.g., MMStar). As suggested by the reviewer, here we perform data deduplication checks between the training set and the evaluation benchmarks. Specifically, we compare image hash values across datasets and compute the number of overlapping instances.
>
> | Deduplication Check | Mathematical | General | Total |
> |:-------------------:|:------------:|:-------:|:-----:|
> |  Overlap Ratio (%)  |       0      |    0    |   0   |
>
> As shown in the table, both the mathematical benchmarks (6 in total) and the general-purpose benchmarks (4 in total) exhibit 0% overlap with the training dataset based on image-hash comparison. This indicates that no duplicate samples exist between the training and evaluation sets, and thus there is no risk of data contamination. In the revised paper, we have included this experiment in Appendix B.17 (highlighted in blue).

---

> ### Author Response · Authors · 2025-11-21
> **Response to Reviewer j11z by Authors (3/8)**
>
> >**W4: The accuracy gate of perception reward**
>
> In training, we incorporate an accuracy gate to prevent reward hacking, where the model might pursue perception rewards at the expense of task accuracy by producing trivially short reasoning. Although the accuracy gate is triggered more frequently on easy problems, resulting in a larger portion of trajectories receiving perception rewards, these rewards are ultimately normalized when converted into advantages. This normalization step ensures that the generally higher rewards obtained from easy examples do not disproportionately influence the final advantage. In contrast, for hard problems, normalization scales up the advantages of correct trajectories, encouraging the model to pay additional attention to these challenging cases. Therefore, the accuracy gate does not bias training toward simpler examples.
>
> To further analyze how the accuracy gate influences the training dynamics of easy and hard problems, as suggested by the reviewer, here we report the accuracy gate activation rates over the course of training for both categories. We classify problems as easy or hard based on the model’s initial accuracy gate activation rate (>0.4 for easy and <0.4 for hard) and track how these activation rates evolve at different training steps. Due to time constraints, we conduct training on a 5000 example subset of the full training set.
>
> |     Training Steps     |   0  |  25  |  50 |  75 |  100 |
> |:----------------------:|:----:|:----:|:----:|:----:|:----:|
> | Activation Rate (Easy) | 0.53 | 0.65 | 0.72 | 0.76 | 0.79 |
> | Activation Rate (Hard) | 0.21 | 0.44 | 0.56 | 0.61 | 0.64 |
>
> As shown in the table, the activation rate increases substantially for both easy and hard questions, and the gap between the two progressively narrows. This further indicates that the model does not over or under-optimize either category during training, suggesting balanced learning dynamics across problem difficulties.
>
> Moreover, to investigate whether a relaxed accuracy gate can further improve performance, we experiment with a curriculum gating strategy, as suggested by the reviewer. Specifically, unlike our default setting, which applies a strict binary gate, we gradually loosen this constraint over the course of training. That is, we progressively assign an increasing proportion of the perception reward to incorrect trajectories and eventually remove the accuracy gate entirely by granting full perception rewards. As before, due to time constraints, we conduct this study using a 5000 example subset of the full training set.
>
> | Method              | MathVerse | LogicVista | Geometry3k |   MMMU   |  MMStar  | HallBench |   Avg.   |
> |---------------------|:---------:|:----------:|:----------:|:--------:|:--------:|:---------:|:--------:|
> | GRPO                |    44.2   |    44.4    |    44.5    |   53.6   |   55.2   |    51.2   |   49.0   |
> | VAPO (strict gate)  |  **48.9** |    47.3    |  **47.7**  | **56.7** |   59.1   |  **55.5** | **52.5** |
> | VAPO (relaxed gate) |    48.5   |  **47.4**  |    47.2    |   56.3   | **59.4** |    55.0   |   52.3   |
>
> As shown in the table, using a relaxed accuracy gate does not yield noticeable performance improvements. This aligns with our earlier analysis that the default accuracy gate does not under-optimize hard examples, and that relaxing the gate introduces a greater risk of reward hacking. We have included this experiment in Appendix B.18 of the revised version (highlighted in blue).

---

> ### Author Response · Authors · 2025-11-21
> **Response to Reviewer j11z by Authors (4/8)**
>
> >**W5: The anchor placement strategy of our approach**
>
> Our current anchor placement strategy resembles the early decision scheme, where anchors are inserted at semantic boundaries (e.g., commas, periods, line breaks) randomly throughout the entire reasoning trajectory. In designing this component, we also considered several alternative placement strategies: 1) *uniform*, where anchors are inserted at fixed intervals rather than randomly; 2) *back-loaded*, where anchors are inserted only in the latter 60% of the reasoning process, acting as a targeted mechanism to address severe visual forgetting that tends to occur toward the end of long trajectories; 3) *front-loaded*, where anchors are inserted only within the first 60% of the trajectory, serving as a baseline comparison. We ensure that all three variants insert the same number of anchors (K=20). We train each variant on a 5000 example subset of the full training set, and we report the results of these placement strategies below.
>
> |   Position   | MathVerse | LogicVista | Geometry3k |   MMMU   |  MMStar  | HallBench |   Avg.   |
> |:------------:|:---------:|:----------:|:----------:|:--------:|:--------:|:---------:|:--------:|
> |     GRPO     |    44.2   |    44.4    |    44.5    |   53.6   |   55.2   |    51.2   |   49.0   |
> |    Uniform   |    48.4   |    47.1    |    47.5    |   56.1   |   58.4   |    55.0   |   52.1   |
> | Front-loaded |    48.1   |    46.7    |    47.0    |   55.9   |   58.5   |    54.4   |   51.8   |
> |  Back-loaded |    48.7   |    46.9    |    47.7    |   56.2   |   59.0   |  **55.8** |   52.4   |
> |    Random    |  **48.9** |  **47.3**  |  **47.7**  | **56.7** | **59.1** |    55.5   | **52.5**
>
> As shown in the table, the simple naive (random) strategy performs better than other placement variants. We attribute this to several factors: 1) uniform insertion, which places anchors at fixed intervals, is more susceptible to reward hacking, as the model may learn to exhibit high visual dependence only at predetermined positions; 2) front-loaded insertion performs noticeably worse than both random and back-loaded insertion, indicating that later positions in the reasoning trajectory are more critical for improving perceptual grounding; 3) In addition, although back-loaded insertion is slightly weaker than random insertion on average, it achieves comparable or even superior performance on MMStar and HallusionBench. This suggests that vision-heavy benchmarks may benefit from a larger number of anchors or greater weighting at later reasoning steps, which aligns with our late-emphasis design. We have included this experiment in Appendix B.19 of the revised version (highlighted in blue).

---

> ### Author Response · Authors · 2025-11-21
> **Response to Reviewer j11z by Authors (5/8)**
>
> >**W6: The comparison to other RL formulations on perception**
>
> VAPO measures perceptual capability across reasoning trajectories through visual anchors and introduces perception reward to encourage stronger reliance on visual information. In contrast, perception-aware policy optimization (PAPO) [B] promotes visually grounded reasoning by enforcing divergence between responses generated from clean versus corrupted images. The two approaches differ in several key aspects: 1) *Granularity of Supervision*. PAPO enhances visual grounding at the sentence level, applying a global constraint on the entire response. VAPO, however, inserts multiple anchors throughout the reasoning process, enabling segment-level and thus more fine-grained supervision. Our late-emphasis design further targets portions of the reasoning where perceptual degradation is most pronounced, supporting controlled and localized optimization; 2) *Efficiency and Stability*. PAPO requires computing a KL divergence loss between clean and corrupted trajectories and supplements this with double entropy regularization to avoid training collapse. In contrast, VAPO builds directly on GRPO and introduces only a single extra perception reward, achieving substantial gains with minimal computational overhead and improved stability, as shown in Table 10 of the Appendix; 3) *Interpretability*. PAPO’s global optimization leaves the evolution of perceptual capability within the reasoning process opaque. VAPO’s anchor scores, by contrast, allow users to monitor and interpret how the model’s perception changes across different reasoning segments and problem types. This provides valuable diagnostic insights for understanding model behavior and improving transparency.
>
> Here, to assess effectiveness, we compare PAPO and VAPO on the established benchmarks. We use the publicly released 7B checkpoint from PAPO and reproduce the results using greedy decoding, as their paper reports performance using the avg@8 accuracy metric, which is not directly comparable to the metrics adopted in prior work [A, C].
>
> |   Method   | MathVerse | LogicVista | Geometry3k |   MMMU   |  MMStar  | HallBench | Avg. |
> |:----------:|:---------:|:----------:|:----------:|:--------:|:--------:|:---------:|:----:|
> | Base Model |    40.7   |    42.6    |    38.5    |   52.7   |   54.9   |    50.0   | 46.6 |
> |    PAPO    |    50.4   |    47.3    |    49.7    | **60.8** |   61.8   |    55.6   | 54.2 |
> |    VAPO    |  **53.3** |  **50.9**  |  **51.3**  |   60.2   | **63.0** |  **57.4** | **56.0** |
>
> As shown in the table, VAPO surpasses PAPO on 5 out of 6 representative benchmarks, achieving an average improvement of around 2% (54.2% → 56.0%). This demonstrates the advantage of VAPO’s vision-anchored, local optimization over PAPO’s sentence-level, global objective, and highlights the importance of our fine-grained late-emphasis design in effectively addressing visual forgetting within the reasoning process. We have included this comparison in Appendix B.20 of the revised version (highlighted in blue).
>
> >**W7: The attention metric**
>
> We follow prior work [D, E] in using an attention-based metric as an indicator of how the model’s reliance on visual information evolves across the reasoning trajectory. We emphasize that this metric is employed purely as a descriptive probe and is not intended to serve as the sole piece of evidence for visual forgetting. Importantly, our core claims regarding visual forgetting and visually grounded reasoning are supported by multiple independent sources of evidence beyond attention: 1) Outcome-level analysis via early decision and error categories. We show that perception errors are the dominant failure mode and that many of these errors become recoverable when the reasoning chain is truncated. This indicates that perceptual degradation is closely tied to increasing reasoning depth; 2) Intervention-based evidence. Visual replay and focus prompts causally modify how often the model looks at the image and consistently reduce late-stage accuracy drop, which provides empirical evidence on visual forgetting; 3) Method validation. Our training signal based on verification of visual claims at anchor points substantially improves model performance, indirectly underscoring the severity of perception degradation.

---

> ### Author Response · Authors · 2025-11-21
> **Response to Reviewer j11z by Authors (6/8)**
>
> >**W8: Related works about reasoning in VLMs**
>
> We thank the reviewer for highlighting this relevant work. Think or Not (TON) [F] is a great and interesting work which primarily investigates when a model should invoke explicit reasoning in order to reduce completion length while preserving accuracy. In contrast, rather than deciding whether to reason, we focus on making the model’s reasoning more effective by mitigating visual forgetting. Thus, TON and our method address complementary questions. TON studies efficiency and selective reasoning, whereas our work examines how to prevent reasoning-induced degradation of visual grounding, thereby unlocking the full potential of multimodal reasoning. We have cited this work in Section 2 of the revised version (highlighted in blue).
>
> >**Q1: Claim quality and robustness**
>
> **Human audit of claim correctness and vision-dependency**: please refer to W1 for human and model evaluation.
>
> **How sensitive is VAPO to weaker generators**: please refer to W1 for other open-source, weaker generators.
>
> **Claim filtering and diversity constraints**: We thank the reviewer for the insightful question. In our current pipeline, we employ a rule-based filtering mechanism and dynamically generate visual claims. That is, for each training example, any claim that fails to meet the required criteria is discarded and resampled until the target number of valid claims is reached. The filtering process incorporates multiple constraints, as detailed in the prompt specification in Fig. 10 of the Appendix, including limits on claim length, the prohibition of ambiguous expressions (e.g., “maybe,” “appears”), and the requirement to maintain a balanced ratio of correct and incorrect claims. To reduce redundancy, we compute pairwise embedding distances between generated claims using Qwen3-Embedding-0.6B, and remove claims with excessively high semantic similarity (> 0.95). In fact, we allow straightforward low-level claims such as color naming, while we impose a mild diversity constraint that limits the number of purely low-level color or counting statements per image through regular expression, favoring a richer distribution of relational, text-based, and global scene-level claims. We will release the full dataset including all generated claims and the complete claim generation scripts upon publication. We have included this claim filtering pipeline in Appendix A.6 of the revised paper.
>
> >**Q2: Reward design**
>
> **Accuracy gate and training bias**: Please refer to W4 for the accuracy gate activation rate over steps and the easy/hard examples traing dynamics.
>
> **Curriculum gating**: Please refer to W4 for the ablation of relaxed gating.
>
> **per-claim calibrated rewards or multi-class verification**: The central contribution of our work lies in introducing visual anchors and corresponding perception reward, which together provide an effective mechanism for assessing and preserving perceptual capability during training. Following the design of GRPO, our reward formulation intentionally remains simple, using easily verifiable, binary judgments to ensure stability and efficiency. While we agree that more calibrated reward formulations, such as multi-class or continuous (confidence-weighted) reward, may offer finer-grained training signals, these extensions are orthogonal to our main contribution of diagnosing visual forgetting and showing that even simple binary anchor rewards already yield consistent gains over a GRPO baseline. We consider exploring richer reward designs, including confidence-weighted or multi-class verification, to be promising directions for future work.

---

> ### Author Response · Authors · 2025-11-21
> **Response to Reviewer j11z by Authors (7/8)**
>
> >**Q3: Anchor placement**
>
> **The impact of anchor position selection to performance**: Please refer to W5 for the exploration of different anchor placement strategies.
>
> **The number of anchors K**: Visual anchors are inserted only during training to compute the perception reward and encourage higher visual reliance. At test time, our inference procedure is identical to GRPO. Consequently, the hyperparameter K affects only the training phase.
>
> Our ablation study in Fig. 7 shows that increasing K (i.e., using more anchors) generally yields positive gains across examples. To further examine whether longer-thought examples benefit disproportionately from larger K, we partition benchmark examples into two groups based on the output length of our VAPO-Thinker-7B model: short-thought (<300 tokens) and long-thought (>300 tokens). We then evaluate models trained with different values of K and analyze how performance trends vary across these two categories.
>
> | Length |   1  |   5  |  10  |  15  |  20  |
> |:------:|:----:|:----:|:----:|:----:|:----:|
> |  Short | 53.7 | 56.0 | 57.5 | 58.4 | 58.5 |
> |  Long  | 47.3 | 49.4 | 50.9 | 51.5 | 51.8 |
> |  Total | 51.5 | 53.8 | 55.1 | 55.8 | 55.9 |
>
> As shown in the table above, increasing K leads to stable performance improvements for both short- and long-thought examples, and the magnitude of these gains remains largely consistent across the two groups. This trend is also aligned with the ablation results reported in Fig. 7 of the main paper. In the revised paper, we have included this analysis in Appendix B.21 (highlighted in blue).
>
> >**Q4: Evaluation rigor**
>
> **Data deduplication checks**: please refer to W3 for the evaluation of the overlapping between training and evaluation set.
>
> **Single-cut recoverable metric**: Please refer to W2 for single-cut position recoverable ratio.
>
> **Reasoning length distribution**: The goal of VAPO is to enhance the effectiveness of reasoning by improving the model’s perceptual capability, rather than to alter the length of its reasoning traces. Accordingly, our pipeline does not incorporate any explicit length regularization. As suggested by the reviewer, we report below the average reasoning length of our models compared with other baselines across the established benchmarks.
>
> |    Models    | MathVerse | LogicVista | Geometry3k | MMMU | MMStar | HallBench | Avg. |
> |:------------:|:---------:|:----------:|:----------:|:----:|:------:|:---------:|:----:|
> | R1-OneVision |    299    |     308    |     287    |  263 |   232  |    201    |  265 |
> |   Vision-R1  |    337    |     293    |     341    |  283 |   239  |    251    |  291 |
> | VAPO-Thinker |    315    |     287    |     334    |  275 |   254  |    238    |  284 |
>
> As shown in the table, the average output length of our model does not differ noticeably from that of other baselines. This rules out the possibility that our performance gains stem merely from longer reasoning traces, and instead indicates that VAPO improves the effectiveness of the reasoning process itself. We have included this experiment in Appendix B.22 of the revised paper (highlighted in blue).

---

> ### Author Response · Authors · 2025-11-21
> **Response to Reviewer j11z by Authors (8/8)**
>
> >**Q5: Comparisons and scope**
>
> **The difference from PAPO**: Please refer to W6 for the conceptual and performance comparison between our method and PAPO. PAPO does not introduce a new reward; instead, it imposes an additional KL term to increase the divergence between responses generated from clean versus corrupted images. This mechanism is compatible with our approach: PAPO can serve as a form of global regularization that modulates sentence-level visual grounding, while our visual anchor–based perception reward may further provide fine-grained, segment-level optimization throughout the reasoning process. Due to time constraints, we leave the exploration of combining these two complementary strategies for future work.
>
> **Multi-image setting**: Our current visual claims are primarily designed for single-image inputs, which is sufficient to validate the core motivation of VAPO, namely improving the model’s reliance on visual information and mitigating visual forgetting. As suggested by the reviewer, we further conduct a small-scale pilot study on multi-image tasks. Specifically, we train VAPO exclusively on the multi-image subset of ViRL39K, which contains approximately 2,500 examples. For claim generation, we consider two types of claims. The first type is single-image claims, which mirror our original setup and are generated with respect to only one image in the multi-image set. The second type is cross-image claims, which capture relationships or distinctions across images. For the former, we provide GPT with only one image at a time. For the latter, we present all images simultaneously and extend the original instructions with an additional requirement that the generated claim must reference information across images (e.g., “the two images depict the same building”). We ensure that each training example contains a balanced number of claims from both categories. The training procedure remains identical to our main pipeline, and GRPO is used as the baseline for comparison. We evaluate the trained models on two representative multi-image benchmarks, including BLINK and MuirBench.
>
> |   Method   |   BLINK  | MuirBench |   Avg.   |
> |:----------:|:--------:|:---------:|:--------:|
> | Base model |   55.3   |    58.1   |   56.7   |
> |    GRPO    |   57.0   |    60.4   |   58.7   |
> |    VAPO    | **60.1** |  **62.2** | **61.2** |
>
> As shown in the table, despite being trained on a limited amount of data, our method still achieves an average improvement of 2.5% over GRPO on multi-image tasks (58.7% -> 61.2%). This demonstrates that our approach is not restricted to the single-image setting and can generalize to multiple images as well. Due to time constraints, we leave the evaluation of video-based tasks as promising future work. We have included this experiment in Appendix B.23 of the revised version (highlighted in blue).
>
> **Reference**
>
> [A] VL-Rethinker: Incentivizing Self-Reflection of Vision-Language Models with Reinforcement Learning, NeurIPS2025
>
> [B] Perception-Aware Policy Optimization for Multimodal Reasoning, arXiv
>
> [C] Vision-R1: Incentivizing Reasoning Capability in Multimodal Large Language Models, arXiv
>
> [D] An Image is Worth 1/2 Tokens After Layer 2: Plug-and-Play Inference Acceleration for Large Vision-Language Models, ECCV2024
>
> [E] Visionary-R1: Mitigating Shortcuts in Visual Reasoning with Reinforcement Learning, arXiv
>
> [F] Think or Not? Selective Reasoning via Reinforcement Learning for Vision-Language Models, NeurIPS2025

---

> ### Author Response · Authors · 2025-11-28
> **Awaiting Your Valuable Feedback**
>
> Dear reviewer j11z,
>
> Thank you for your thoughtful comments and for acknowledging the contributions of our work. Your feedback has greatly helped us refine the paper and enhance its clarity. We have provided detailed responses and corresponding revisions to address each of your concerns. As the discussion period is nearing its end, we would appreciate hearing whether our updates sufficiently resolve the issues you raised. Please feel free to share any additional questions or suggestions.
>
> Best regards,
>
> Authors

---

### Official Review · Reviewer_KfZR · 2025-10-31

**Soundness:** 1
**Presentation:** 2
**Contribution:** 2
**Rating:** 4
**Confidence:** 5

**Summary:**

This paper investigates a "dual nature" of reasoning in Vision-Language Models (VLMs), claiming that while reasoning enhances logical inference, it can impair perceptual grounding. The authors attribute this phenomenon to "visual forgetting," where prolonged reasoning causes the model to disregard visual input. They propose VISION-ANCHORED POLICY OPTIMIZATION (VAPO), an RL-based method with a "perception reward" to force the model to maintain visual grounding

**Strengths:**

1. The "early decision" mode of analysis (Fig. 2A) is a clever and effective method for demonstrating this phenomenon of the degradation of perceptual accuracy as the generation process continues.

2. VAPO showcase an effective solution of rewarding the model for maintaining perceptual grounding,

**Weaknesses:**

The primary weakness of this paper lies in its central conceptual claim. The argument that "reasoning harms perception" is a significant overstatement that is neither logically sound nor intuitively convincing.

1. **Flawed Causal Claim (Logical):** The evidence presented demonstrates a **correlation between reasoning length and perceptual decline, but it fails to establish causation.** The "prolonged reasoning" is perfectly confounded with the length of the generated text context. The paper provides no evidence to disprove the more parsimonious hypothesis: any sufficiently long context (it could be textual reasoning or it could be many image tokens) would cause the model's attention to "drift" from the visual input, leading to perceptual errors. The paper mistakes correlation for causation.

2. **Flawed Premise (Intuitive):** The claim is also nonsensical from a cognitive perspective. In human intelligence, perception and reasoning are synergistic, not conflicting. Perception provides the "ground truth" (premises) upon which reasoning operates. Reasoning, in turn, can direct perceptual focus (e.g., "look for the red car"). Claiming an inherent conflict in AI models based on this confounded evidence is an extraordinary claim that would require extraordinary proof, which the paper does not provide. It is far more likely that the model is simply failing at a known, mundane problem (long-context attention drift) than it has developed a special cognitive conflict different from how human visual system and brain work.

The main experiment results do not convincingly show the significance of perceptual grounding, without comparing with state-of-the-art RL approaches on VLM that do not explicitly facilitate perceptual grounding, such as MM-Eureka and VL-Rethinker.

Since the central argument is flawed and misleading, my true rating for this paper is 3. From my perspective, the current version is not ready for publication.

**Questions:**

1. On the Confounding Variable: Did the authors run any control experiments to disentangle the act of reasoning from the length of the textual context? For example, what happens to perceptual accuracy if the model is fed a long context of another question-solution or a few other images ?
2. On the Experimental Design: The "visual replay" intervention (Fig. 3) simply re-inserts the image. This intervention successfully alleviates the performance drop. Does this not strongly suggest that the problem is a standard attention failure (i.e., the model "forgot" the image in its context) rather than a more complex cognitive "harm" caused by the reasoning process itself?
3. On the Method's Motivation: Given the lack of evidence for a true "reasoning vs. perception" conflict, isn't VAPO more accurately described as a powerful regularization technique to combat visual context drifting? Why frame it as solving a cognitive duality when the evidence only points to a context length/attention failure?
4. On the Error Analysis: In Fig. 2, the paper claims reasoning induces perception errors. Could the causality be reversed? That is, could the model make an early logical error (e.g., path 1 in Fig 1, deciding 7.4+7.2+8.1=22.7 instead of 20.5) and then hallucinate visual "evidence" to justify its flawed reasoning path, rather than the reasoning itself degrading its ability to see?

---

> ### Author Response · Authors · 2025-11-21
> **Response to Reviewer KfZR by Authors (1/3)**
>
> We would like to thank the reviewer for their constructive comments and questions. Following are our responses to your concerns.
>
> **"reasoning harms perception" is an overstatement**: We respectfully disagree with the reviewer that our finding is an overstatement. In our paper, we state the original finding that "while reasoning substantially enhances logical inference and facilitates performance on challenging problems, it may gradually impair perceptual grounding". It is crucial to clarify that this statement expresses a possibility, which is explicitly conditioned on the length of the reasoning process. As illustrated in Fig. 2, our early-decision analysis reveals the effect of reasoning length on accuracy, which exhibits a characteristic pattern of increase, saturation, and eventual decline. This trend reflects a subtle tension between logical inference ability and perceptual capability as the reasoning chain grows longer. In the early phase, the degradation of perceptual capability is negligible, allowing longer reasoning to improve overall performance. In the later phase, however, the accumulated impairment of perception counteracts and ultimately outweighs the benefits brought by extended reasoning, leading to performance saturation and degradation. **Therefore, reducing our finding to a blanket statement "reasoning harms perception" without the conditions would be inaccurate and misleading.**
>
> >**W1: Our claim in a logical sense**
>
> We respectfully disagree with the reviewer’s assertion that our analysis demonstrates only correlation rather than causation. First, regarding the hypothesis suggested by the reviewer "any sufficiently long context (it could be textual reasoning or it could be many image tokens) would cause the model's attention to "drift" from the visual input, leading to perceptual errors", We have already presented sufficient counterexamples. In Section 3.2 of the main paper, we show through the focus prompt intervention that inserting specific textual contexts such as explicit instructions into the reasoning process can substantially mitigate visual forgetting (i.e., the model’s attention drifting away from the visual input, as noted by the reviewer), thereby disproving the hypothesis that any sufficiently long context necessarily induces visual forgetting. Furthermore, our visual replay intervention demonstrates that inserting image tokens associated with the query similarly alleviates perceptual degradation, contradicting the hypothesis that any long contexts, including those containing image tokens, inherently cause visual forgetting. **Together, these findings provide strong evidence that the reviewer’s hypothesis does not hold.**
>
> Besides, in our paper, we attribute the observed decline in perceptual capability with increasing reasoning length to visual forgetting. However, we did not claim that visual forgetting occurs exclusively during reasoning. In fact, as discussed in Section 2, we reviewed prior work on visual forgetting and highlighted that the phenomenon has become increasingly prominent as modern VLMs operate with ever-longer output contexts. Reasoning constitutes a particularly common subset of long context, and thus naturally and significantly amplifies visual forgetting. Furthermore, our multiple interventions including early decision, visual replay, and focus prompt, collectively demonstrate the fact that prolonged reasoning may lead to perceptual degradation.

---

> ### Author Response · Authors · 2025-11-21
> **Response to Reviewer KfZR by Authors (2/3)**
>
> >**W2: Our claim in a intuitive sense**
>
> **We would first like to emphasize that VLMs are computational models rather than cognitive models, and therefore drawing conclusions based on notions of human sensibility or brain-like processing is not reliable.** Moreover, our main contribution lies in identifying a tension between reasoning depth and perceptual fidelity in existing VLMs arising from visual forgetting. We did not claim that there is an inherent conflict between reasoning and perception in general visual systems. In fact, Fig. 2 of the main paper shows that increasing reasoning depth is beneficial within a certain range, and only after surpassing a particular threshold does performance saturate and eventually decline. This pattern indicates that reasoning and perception mostly exhibit synergistic effects for computational models, and the degradation observed at larger depths stems from accumulated visual forgetting rather than an intrinsic incompatibility between the two. Moreover, by mitigating visual forgetting with our proposed VAPO method, Fig. 5(B) demonstrates that accuracy continues to improve steadily as reasoning unfolds. This behavior shows that the successfully recovered model exhibits the desired strong synergy between perception and reasoning by maintaining visually grounded reasoning throughout the entire process.
>
> **Comparison with more RL approaches**: In our experiments, we compare VAPO-Thinker-7B against strong baseline models of comparable parameter scale that are trained with GRPO or its close variants, most of which do not explicitly emphasize perceptual grounding. Our method achieves average improvements of 2% and 3.2% on mathematical and general-purpose benchmarks, respectively, with particularly notable gains on MMStar and HallusionBench. These results highlight the effectiveness of VAPO on visual tasks, especially those that are perception-heavy. As suggested by the reviewer, here we additionally include two baselines, MM-Eureka [A] and VL-Rethinker [B]. We use their publicly released 7B checkpoints and reference the reported results in their papers, reproducing the numbers when necessary.
>
> |    Method    | MathVerse | LogicVista | Geometry3k |   MMMU   |  MMStar  | HallBench |   Avg.   |
> |:------------:|:---------:|:----------:|:----------:|:--------:|:--------:|:---------:|:--------:|
> |   MM-Eureka  |    50.3   |    48.5    |    50.2    |   55.7   |   60.4   |    54.8   |   53.3   |
> | VL-Rethinker |  **54.2** |    47.2    |    49.6    |   57.9   |   61.8   |    55.4   |   54.4   |
> | VAPO-Thinker |    53.3   |  **50.9**  |  **51.3**  | **60.2** | **63.0** |  **57.4** | **56.0** |
>
> As shown in the table, our model achieves state-of-the-art performance on 5 of the 6 benchmarks, surpassing the previous best results by an average of 1.6% (54.4% -> 56.0%). This further demonstrates that our approach, by producing more visually grounded reasoning, substantially enhances the effectiveness of reasoning across a diverse range of tasks. We have included this additional comparison in Appendix B.20 of the updated paper (highlighted in blue).

---

> ### Author Response · Authors · 2025-11-21
> **Response to Reviewer KfZR by Authors (3/3)**
>
> >**Q1: Confounding Variable**
>
> As noted in W1, we have provided sufficient evidence that longer textual or image contexts do not necessarily induce visual forgetting through focus prompt and visual replay, respectively. Moreover, we did not claim that visual forgetting occurs exclusively during reasoning. Rather, our emphasis is that reasoning, as a particularly common and often lengthy form of context, tends to exacerbate visual forgetting prominently.
>
> >**Q2: Experimental Design**
>
> As discussed in W1 and W2, we have never framed the issue as a form of cognitive harm nor as an inherent conflict between reasoning and perception in the paper. Instead, we consistently attribute the phenomenon to visual forgetting induced by extended reasoning length.
>
> >**Q3: Method Motivation**
>
> In the paper, we describe our method as “a simple yet effective algorithm that explicitly guides the reasoning process toward a visually grounded trajectory”. This phrasing highlights that VAPO is designed to increase a model’s reliance on visual information during reasoning, as our goal is to identify the perceptual gap in current multimodal reasoning systems and improve their performance. From a broader perspective, as the reviewer suggested, the same mechanism could also be applied to mitigate visual drifting in other contexts beyond the reasoning setting. The duality of reasoning in our formulation refers to its benefits and drawbacks with respect to effectiveness, rather than cognitive conflicts between reasoning and perception. Specifically, reasoning can provide strong logical support, yet its extended length can also induce visual forgetting. VAPO resolves this duality by alleviating the visual degradation caused by long reasoning traces, thereby allowing reasoning to retain its benefits without incurring the accompanying perceptual decline.
>
> >**Q4: Error Analysis**
>
> We thank the reviewer for the insightful question. In our error analysis, based on the recorded human audit of 500 randomly sampled failure cases, we found that instances containing multiple error types within the same response are quite rare, occurring in fewer than 20 examples (<4%), which is also the reason we prompt GPT to directly identify the primary error category as shown in Fig. 8 of the Appendix. To further investigate this, we manually re-examine these cases and record the specific ordering patterns in which different error categories co-occur.
>
> | Error Category | Logical -> Perception | Perception -> Logical | Knowledge -> Perception | Knowledge -> Logical |
> |:--------------:|:---------------------:|:---------------------:|:-----------------------:|:--------------------:|
> |   # Examples   |           5           |           7           |            3            |           1          |
>
> As shown in the table, any error category combinations not listed above did not occur in our samples. Notably, cases in which logical errors precede perceptual errors are extremely rare. This rarity ensures that such patterns do not meaningfully affect our experimental results or conclusions. It also makes it difficult to draw any substantive connection between perceptual errors and other error types, given how infrequently they co-occur in practice. These results further reinforce our finding that longer reasoning contexts tend to induce visual forgetting, which in turn leads to the observed performance degradation. In the updated version, we have included this analysis in Appendix A.3 (highlighted in blue).
>
> **Reference**
>
> [A] MM-Eureka: Exploring the Frontiers of Multimodal Reasoning with Rule-based Reinforcement Learning, arXiv
>
> [B] VL-Rethinker: Incentivizing Self-Reflection of Vision-Language Models with Reinforcement Learning, NeurIPS2025

---

> ### Author Response · Authors · 2025-11-28
> **Awaiting Your Valuable Feedback**
>
> Dear reviewer KfZR,
>
> Thank you very much for the time and care you invested in reviewing our submission. Your comments have been highly valuable in strengthening the technical rigor of the paper. We have carefully addressed each of your points through detailed responses and corresponding revisions. With the discussion phase drawing to a close, we would appreciate hearing whether these updates resolve the concerns you raised. Please feel free to let us know if any additional clarifications or suggestions would be helpful.
>
> Best regards,
>
> Authors

---

> ### Comment · Reviewer_KfZR · 2025-11-28
>
> Thank the authors for the rebuttal.
>
> **I am currently re-evaluating the paper and considering raising the score. If the authors address the below requests, I will do so.**
>
> First, I am glad to see that new baseline results are added, and I would like to see them properly cited in the revision.
>
> However, there is a question lingering and I hope the authors to clarify.
>
> The evidences from the rebuttal support the root cause of the "visual forgetting": longer sequence of textual tokens moves the context window away from image tokens, so the model lose its focus on the images and the accuracy decreases.
>
> The author's claim on "reasoning impairs perceptual grounding" maintains my core concern.
> My question is:
> - does longer sequence of textual tokens equal reasoning?
> - Are they really doing reasoning or they are just pretend to do so because the base model has been taught so?
> The evidence of "Early Decision" experiments show that models often possess the correct answer initially but drift into error as they continue generating. **Ain't this support that the model is just pretending to do the reasoning, outputing a reasoning format?**
>
> **I think the authors might mistake "the behavior of reasoning", the longer output of textual tokens, with "the genuine reasoning", and this mistake explains the counter-intuition.**
>
> I still don't believe the fancy statement of "reasoning harms perception". If so, it is suggesting that perception and reasoning can only trade-off for each other in vision-language reasoning, instead of mutually help?
>
> I am prepared to raise the score if the authors address my concerns.

---

> ### Author Response · Authors · 2025-11-28
> **Follow-up Response by Authors**
>
> We would like to sincerely express our gratitude once again for your response and thoughtful consideration to our work. Below, we provide additional clarifications addressing your question and concern.
>
> >**New baseline citations**
>
> We thank the reviewer for highlighting these valuable related works. We have now cited the suggested references in Section 2 of the revised paper (highlighted in blue).
>
> >**Q1: does longer sequence of textual tokens equal reasoning?**
>
> Not quite. As noted in our earlier response, **textual context can encompass many types of input such as questions, documents, and more, not only reasoning.** Longer contexts may lead to visual forgetting, and reasoning, as one of the most common and typically lengthier forms of context, tends to intensify visual forgetting more significantly.
>
> >**Q2: Are they really doing reasoning or they are just pretend to do so?**
>
> We believe that the model is performing effective reasoning. It is important to note that **although the longer generated reasoning context may introduce errors due to visual forgetting as the reviewer suggested, the overall effect of reasoning on model performance remains strongly positive.** As shown in Fig. 2(A) of the main paper, accuracy as a function of reasoning length follows an increase–saturation–decrease pattern. Although visual forgetting introduces some degradation in the later steps, the initial gains remain dominant, indicating that the full reasoning trace is still beneficial overall. Our work aims to further enhance the effectiveness of reasoning by mitigating the errors introduced by late-stage visual forgetting, and our empirical results demonstrate that the proposed approach successfully achieves this.
>
> >**The concern about "reasoning harms perception"**
>
> We understand the reviewer’s concern. The term "reasoning" can indeed cover multiple aspects, including reasoning length and reasoning behavior, as the reviewer noted. **To clarify, in our claim that “longer reasoning may gradually impair perceptual grounding”, the observed perceptual decline stems fundamentally from reasoning length instead of reasoning behavior**. We did not mistake or confuse “the behavior of reasoning” and “longer textual outputs” in the paper. In fact, as mentioned in our earlier response, Section 2 explicitly acknowledges that the origin of visual forgetting lies in context length, and reasoning, as a particular instance of long-context generation, naturally inherits this issue, which may limit its effectiveness. **To improve clarity in the paper, we have slightly rephrased our main finding in the revised version (highlighted in blue), explicitly emphasizing that it is the length of reasoning, rather than reasoning behavior itself, that affects perceptual grounding.**
>
> >**The trade-off between reasoning and perception**
>
> As mentioned in our previous response, **the trade-off between reasoning and perception arises only when the reasoning length becomes sufficiently large, which can lead to visual forgetting.** Within a moderate range of reasoning length as shown in Fig. 2(A), accuracy increases steadily, indicating a mutually beneficial interaction between reasoning and perception. This trade-off induced by excessively long reasoning length is not an inherent property of VLMs but rather a consequence of training limitations. Our method effectively mitigates this issue; as shown in Fig. 5(B), the accuracy of our model continues to improve with longer reasoning, without exhibiting a noticeable trade-off.
>
> We once again thank you for your prompt response and careful evaluation of our work. Your comments have significantly strengthened the clarity and rigor of the paper. We hope that our additional responses further address your remaining concerns.

---

> > ### Comment · Reviewer_KfZR · 2025-11-28
> >
> > I would like to thank the authors for their patience and thoughtful responses. Based on the revisions, I have raised my score to 6 (note: I am unable to update this in the system as it is now locked).
> >
> > I agree that the paper offers valuable insights to the community, and I am pleased to see that the authors have adjusted their tone regarding the claim that "reasoning harms perception." I hope the authors understand that my concern is not about being overly critical of this particular paper. Rather, it is a general caution that authors should exercise care when making strong claims, especially those that may be counter-intuitive to many readers and could potentially mislead junior researchers. It is thus recommended that the authors carefully re-consider (and then clearly state) in the paper about the premises underlying the claims or hypotheses for explaining their observations.

---

> > > ### Author Response · Authors · 2025-11-28
> > > **Thank You for Your Feedback**
> > >
> > > Dear reviewer KfZR
> > >
> > > We are sincerely grateful for your response and happy to know that our clarifications have effectively addressed your concerns. We also agree that clear and precise claims are valuable for the broader community of readers and researchers, and your suggestions have been crucial in improving the quality of our paper.
> > >
> > > We also fully understand the current system issue preventing you from updating the score. We trust that the AC will be aware of this, and we expect the issue will be resolved soon. we would greatly appreciate it if you could update the score after the system is fixed.
> > >
> > > Thank you once again for the time and effort you devoted to reviewing our paper.
> > >
> > > Best regards,
> > >
> > > Authors

---

### Official Review · Reviewer_4nqT · 2025-11-01

**Soundness:** 4
**Presentation:** 3
**Contribution:** 3
**Rating:** 6
**Confidence:** 3

**Summary:**

This paper investigates a key contradiction in Vision-Language Models (VLMs), termed the “dual nature” problem. While Chain-of-Thought (CoT) reasoning improves logical inference, excessively long reasoning chains can impair visual grounding, leading to errors on simple visual tasks. The authors attribute this to “visual forgetting,” where attention to the image fades as reasoning progresses.
To validate this, the paper introduces two inference-time interventions:
1. Visual Replay – reintroducing the image mid-reasoning.
2. Focus Prompt – instructing the model via text to re-attend to the image.
 Both methods mitigate degradation, supporting the visual forgetting hypothesis.
To address the issue fundamentally, the authors propose VISION-ANCHORED POLICY OPTIMIZATION (VAPO), an RL-based algorithm that inserts visual anchors (short, image-related statements) into reasoning paths. The model judges their correctness and receives a perception reward, encouraging sustained visual grounding.

**Strengths:**

1. Important and Novel Problem: Identifies a counterintuitive phenomenon—“more reasoning can reduce accuracy.” This challenges a major assumption in VLM research.
2. Strong Empirical Validation: The “visual forgetting” hypothesis is convincingly supported through attention visualizations and well-designed interventions (Visual Replay, Focus Prompt).
3. Elegant Method Design: VAPO elegantly integrates perception evaluation into RL training through visual anchors and perception rewards, directly addressing the root cause.
4. Comprehensive Evaluation:
  - Covers 10 benchmarks across reasoning and VQA tasks.
  - Compares against both closed- and open-source SOTA models.
  - Provides detailed analyses and ablations confirming the method’s robustness.

**Weaknesses:**

1. Reproducibility: Reliance on a closed-source, super-capable large model (GPT-5) may create difficulties for other researchers trying to reproduce or extend this work.
2. Cost and Efficiency: Generating a large number of visual claims for the training data introduces considerable computational overhead.
3.  Quality Bottleneck: The upper bound of VAPO's effectiveness may be limited by the quality of the claim generation model. Low-quality claims could negatively impact training.
4. Added Complexity: Introduces additional steps, hyperparameters (K, β, γ), and tuning difficulty compared to standard GRPO.
5. Efficiency Discussion Limited: Although runtime overhead is small (Table 10), total pipeline cost—including data generation—remains higher than baseline.

**Questions:**

1. Sensitivity to Claim Quality: How does performance change if visual claims are generated by weaker models (e.g., LLaVA-1.6, Qwen-VL-Max)? Are there filtering mechanisms?
2. Universality of Visual Forgetting: This study experiments with models based on Qwen2.5-VL. Do you believe the "visual forgetting" phenomenon is prevalent across all types of VLM architectures?
3. Generalization of VAPO: Does the perception training generalize to fine-grained or spatial reasoning tasks beyond truth verification?
4. Alternative Anchors:  Besides using GPT-5 to generate explicit text claims, have you considered other forms of "anchors"? For example, could you explore self-supervised or reconstruction-based anchors to reduce dependence on external models?

**Details Of Ethics Concerns:**

no concerns.

---

> ### Author Response · Authors · 2025-11-21
> **Response to Reviewer 4nqT by Authors (1/4)**
>
> We greatly appreciate your insightful comments and your recognition of our work. Here are our responses to your concerns.
>
> >**W1: The reproducibility of our work**
>
> In Appendix A, we provided comprehensive details on the prompts used for generating visual claims with GPT, including the generation rules, output format, and quantity specifications. We also included illustrative examples of visual claims to offer researchers a clear reference standard. For training, we reported the statistics of both the model and the dataset, along with the sizes and splits of the evaluated benchmarks. Moreover, we described the full training configurations, such as hyperparameters, training framework, and computational specifications to facilitate reproducibility. Finally, we will release our dataset (including all generated claims), the trained model (3B and 7B), and the code for claim generation and model training upon publication.
>
>
> >**W2: The computational efficiency of our method**
>
> At test time, our method follows the same inference pipeline as GRPO and does not introduce extra computational overhead. The time costs of our proposed method can be decomposed into two parts:
>
> - **The cost of training.**
>
> In Appendix B.9, we compared the cost-gain trade-offs of VAPO with several widely used policy optimization algorithms, including GRPO and DAPO. For the reviewer’s convenience, we directly present the relevant results here.
>
> | Method | Epoch |  Time  | Accuracy |  Gain |
> |:------:|:-----:|:------:|:--------:|:-----:|
> |  GRPO  |   2   | 18h46m |   51.54  |   -   |
> |  DAPO  |   2   | 25h11m |   53.17  | +1.63 |
> |  VAPO  |   2   | 19h14m |   55.91  | +4.37 |
>
> As shown, DAPO requires approximately 6 additional hours of training time relative to GRPO while yielding an average accuracy improvement of only 1.63%. In contrast, our method incurs only about 30 minutes of additional training time yet achieves 4.37% improvement over the baseline, demonstrating a substantially better cost-efficiency ratio. We further analyze this advantage by noting that our anchor scoring procedure relies solely on a binary judgement, which is essentially a single-token rollout, making it highly efficient. Moreover, the gradient update process in our method is identical to that of GRPO, introducing no extra computational overhead.
>
> - **The cost of visual claim generation.**
>
> In VAPO, we leverage LLMs, typically GPT to generate visual claims for assessing the model's perception capability. As suggested by the reviewer, we provide below an estimate of the computational time required to generate these visual claims, based on our usage records. Specifically, we employ batch inference following the implementation in [A], allowing multiple queries to be processed concurrently and substantially improving efficiency.
>
> | Dataset | Total | Querying | Filtering |
> |:-------:|:-----:|:----------:|:---------:|
> | ViRL39K | 38min |    34min   |    4min   |
>
> As shown in the table, the time is mainly spent on querying GPT and on the subsequent filtering process. The querying stage refers to the waiting time after submitting prompts, which also includes re-queries for generated claims that fail to meet the specified requirements. The filtering stage applies a set of rule-based criteria (as described in Fig. 10) to remove claims that do not satisfy our standards, for example, those that are excessively long, contain uncertainty markers such as "maybe" or "appears", or violate the required balance between correct and incorrect claims. Besides, we also remove redundant claims that are semantically repetitive with embedding models, and limit the number of claims from various types (e.g., color, relational, OCR) through regular expressions. Any filtered-out examples are resubmitted until the target number of valid claims is reached. Overall, the entire process takes roughly half an hour, which is negligible compared with the training phase and remains highly efficient relative to methods such as DAPO even considering the overall pipeline costs. It is important to note that this claim generation process is a one-time procedure, and no further GPT querying is required for subsequent training runs. We have included this experiment into Appendix B.9 of the revised paper (highlighted in blue).

---

> ### Author Response · Authors · 2025-11-21
> **Response to Reviewer 4nqT by Authors (2/4)**
>
> >**W3: The quality of generated visual claims**
>
> In our main experiments, we employ GPT-5 to generate visual claims due to its strong performance and cost-effectiveness. As suggested by the reviewer, we further examine how using open-source, weaker models to generate visual claims influences VAPO's effectiveness. Specifically, we consider two typical VLMs: LLaVA-1.6-34B and Qwen2.5-VL-32B. We leave Qwen-VL-Max for future work, as it is not fully open-sourced yet and currently provides limited access. For each model, we input the instructions directly as user prompts and apply a filtering mechanism similar to that used with GPT-5 to remove claims that do not meet our requirements. Due to time constraints, we train VAPO on a subset of 5000 examples randomly sampled from the full training set using the claims generated by these models, and evaluate results on the established benchmarks.
>
> |    Model   | MathVerse | LogicVista | Geometry3k | MMMU | MMStar | HallBench | Avg. |
> |:----------:|:---------:|:----------:|:----------:|:----:|:------:|:---------:|:----:|
> |    GRPO    |    44.2   |    44.4    |    44.5    | 53.6 |  55.2  |    51.2   | 49.0 |
> |  LLaVA-1.6 |    48.1   |    46.7    |    47.0    | 56.5 |  58.6  |    55.0   | 52.0 |
> | Qwen2.5-VL |    48.5   |    47.0    |    47.4    | 56.7 |  58.9  |    55.2   | 52.3 |
> |    GPT-5   |    48.9   |    47.3    |    47.7    | 56.7 |  59.1  |    55.5   | 52.5 |
>
> As shown in the table, despite being open-source models, Qwen2.5-VL achieves performance comparable to those obtained with GPT-5, yielding nearly identical average improvements over GRPO (±0.2%). Similarly, LLaVA-1.6, one of the pioneering VLMs, also attains nearly the full performance achieved with GPT-5 (±0.5%). These results indicate that VAPO does not have to rely on closed-source, state-of-the-art models to be effective; smaller open-source models are capable of delivering almost the same performance gains. In the revised version, we have added this experiment to Appendix B.12 (highlighted in blue).
>
> >**W4: The complexity of our method**
>
> Compared with GRPO, our method introduces visual claim generation and perception reward to encourage the model to rely more effectively on visual inputs. However, as demonstrated in the efficiency analysis above, VAPO achieves substantial performance improvements with only a minimal additional time cost, making it significantly more cost-effective than alternative approaches such as DAPO. Regarding hyperparameter tuning, our ablation studies have already identified generally optimal choices for K, β, and γ. Researchers can simply reproduce our configurations or adjust them based on the trends and analyses provided to tailor the settings to their specific use cases.
>
> >**W5: The efficiency discussion of our approach**
>
> Please refer to W2 for comprehensive efficiency analysis and discussion.

---

> ### Author Response · Authors · 2025-11-21
> **Response to Reviewer 4nqT by Authors (3/4)**
>
> >**Q1: Sensitivity to claim quality and filtering mechanism**
>
> Please refer to W2 and W3 for filtering mechanism and sensitivity to claim quality, respectively.
>
> >**Q2: The universality of visual forgetting**
>
> We thank the reviewer for the insightful question. Following prior work [B, C], we use Qwen2.5-VL as a standard base model to evaluate the effectiveness of our method, given its strong and representative reasoning capabilities. We posit that visual forgetting is primarily driven by two factors: 1) Training data. The existing training corpora (e.g., mathematics or text-intensive tasks) place heavy emphasis on textual patterns, leading models to develop a strong text bias; 2) Training objective. The widespread use of next-token prediction objective encourages models to prioritize autoregressive consistency, causing attention to visual inputs to become progressively diluted as the output sequence grows, as shown in Fig. 3 of the main paper. Therefore, unless these underlying factors are substantially altered, we expect that visual forgetting will likely generalize across different model architectures.
>
> To provide a preliminary validation of this assumption, here we additionally experiment with Qwen3-VL-8B, a more recent and more capable reasoning model whose architecture differ substantially from those of Qwen2.5-VL. We apply VAPO to Qwen3-VL to assess whether our method remains effective under different scenarios. Due to time constraints, we leave exploration of other base models for future work.
>
> |   Model  | MathVerse | LogicVista | Geometry3k |   MMMU   |  MMStar  | HallBench |   Avg.   |
> |:--------:|:---------:|:----------:|:----------:|:--------:|:--------:|:---------:|:--------:|
> | Qwen3-VL |    58.8   |    56.7    |    59.3    |   68.4   |   67.3   |    59.2   |   61.8   |
> |  + GRPO  |    63.5   |    60.1    |    64.4    |   72.7   |   68.3   |    60.4   |   64.9   |
> |  + VAPO  |  **67.6** |  **65.8**  |  **68.7**  | **74.2** | **72.5** |  **63.9** | **68.8** |
>
> As shown above, our method continues to provide substantial improvements over GRPO when applied to Qwen3-VL, yielding an average gain of 3.9% (64.9% -> 68.8%). This suggests that, beyond Qwen2.5-VL, other reasoning models may also suffer from pronounced visual forgetting, and that our method is broadly applicable and effective across different model architectures. We have included this experiment in Appendix B.13 of the revised paper (highlighted in blue).
>
> >**Q3: Generalization of VAPO to fine-grained/spatial reasoning tasks**
>
> During training, our visual claims are designed to span multiple dimensions of perceptual capability, including fine-grained visual details (e.g., "the whiteboard says No Parking"), and spatial reasoning (e.g., "the bus is to the left of the motorcycle"). Consequently, our perception training enhances performance on tasks requiring these capabilities. In fact, some of the benchmarks we evaluate on, such as MMStar and HallusionBench, already include subsets of spatial reasoning and fine-grained recognition problems, suggesting that our method exhibits strong generalization. For completeness, here we further evaluate our model on two popular benchmarks, V* [D] and RealWorldQA, which target fine-grained perception and spatial reasoning, respectively.
>
> |     Model    | V* (Fine-grained) | RealWorldQA (Spatial) | Avg. |
> |:------------:|:-----------------:|:----------------------:|:----:|
> |  Qwen2.5-VL  |        71.4       |          64.3          | 67.9 |
> | R1-OneVision |        73.5       |          65.7          | 69.6 |
> | VLAA-Thinker |        74.8       |          66.3          | 70.6 |
> |   Vision-R1  |        74.1       |          66.8          | 70.5 |
> | VAPO-Thinker |        77.9       |          69.4          | 73.7 |
>
> As shown in the table, our model achieves an average improvement of 3.1% (70.6% -> 73.7%) over the previous best results on both fine-grained and spatial reasoning benchmarks. This empirically demonstrates that VAPO generalizes well to a broad range of perception-intensive tasks. We have included this experiment in Appendix B.14 of the revised paper (highlighted in blue).

---

> ### Author Response · Authors · 2025-11-21
> **Response to Reviewer 4nqT by Authors (4/4)**
>
> >**Q4: Alternative anchors besides using textual claims**
>
> Yes, it is possible to consider alternative forms of anchors beyond textual claims. The core idea of VAPO is to utilize anchors to assess model's perception capability during reasoning process and use curated rewards to encourage stronger reliance on visual inputs. The specific form of the anchor (i.e., the mechanism used to measure perception), is not essentially limited to binary textual claims. We choose textual claims mainly because 1) *efficiency*: As discussed in the efficiency analysis, current pipeline introduces only minimal training overhead, offering an excellent cost-performance trade-off; 2) *simplicity*, unlike reconstruction-based methods which may require additional learnable decoders or more complex architectures, textual claims allow us to probe perception capability with only a single-token output; 3) *effectiveness*: Our experiments also demonstrate that textual claims significantly improve model accuracy, surpassing previous strong baselines.
>
> In fact, when designing the visual anchors, we also considered reconstruction-based anchors, where the model would be required to reconstruct visual features or undergo linear probing to assess its perceptual capability. However, as mentioned above, these approaches introduce additional learnable parameters, thereby increasing computational and training complexity.
>
> Another promising self-supervised direction is to adopt a bootstrapping approach, in which the model generates its own visual claims and subsequently uses them as training signals, thereby eliminating the need for external models. Here, as suggested by the reviewer, we explore whether this more efficient training strategy for VAPO remains effective. Specifically, instead of relying on GPT to produce visual claims, we allow the model being trained to generate its own visual claims during the rollout step using the provided instructions, and then use these self-generated claims to assess its perceptual capability and update the model accordingly. This procedure is self-supervised and iterative: the model’s generated claims help improve its visual capability, which in turn further refines the quality of subsequent claims. Due to time constraints, we use Qwen2.5-VL-7B as the base model and train on a 5000 example subset of the full training set.
>
> | Model                  | MathVerse | LogicVista | Geometry3k | MMMU | MMStar | HallBench | Avg. |
> |------------------------|:---------:|:----------:|:----------:|:----:|:------:|:---------:|:----:|
> | Base Model             |    40.7   |    42.6    |    38.5    | 52.7 |  54.9  |    50.0   | 46.6 |
> | GRPO                   |    44.2   |    44.4    |    44.5    | 53.6 |  55.2  |    51.2   | 49.0 |
> | VAPO (Self-Supervised) |    48.1   |    46.5    |    47.4    | 55.9 |  57.9  |    55.1   | 51.8 |
> | VAPO (GPT)             |    48.9   |    47.3    |    47.7    | 56.7 |  59.1  |    55.5   | 52.5 |
>
> As shown in the table, this self-supervised approach retains the vast majority of the effectiveness observed in the default (GPT) setting, with only a modest average difference of 0.7% (46.5% → 47.3%). This indicates that, even without relying on any external or stronger models, VAPO can still achieve highly promising performance. In the revised paper, we have included this experiment in Appendix B.15 (highlighted in blue).
>
>
> **Reference**
>
> [A] Visual Classification via Description from Large Language Models, ICLR2023
>
> [B] Vision-R1: Incentivizing Reasoning Capability in Multimodal Large Language Models, arXiv
>
> [C] VL-Rethinker: Incentivizing Self-Reflection of Vision-Language Models with Reinforcement Learning, NeurIPS2025
>
> [D] Guided Visual Search as a Core Mechanism in Multimodal LLMs, CVPR2024

---

> ### Author Response · Authors · 2025-11-28
> **Awaiting Your Valuable Feedback**
>
> Dear reviewer 4nqT,
>
> We would like to thank you for your insightful feedback and recognition of our work. We believe your comments have been instrumental in improving the comprehensiveness and clarity of our paper. We have presented responses and revisions addressing each listed concern and question. As the discussion deadline approaches, we would like to know if our revisions effectively address your concerns. Please let us know if you have any further questions or suggestions.
>
> Best,
>
> Authors

---

### Official Review · Reviewer_YzHK · 2025-11-04

**Soundness:** 3
**Presentation:** 4
**Contribution:** 3
**Rating:** 6
**Confidence:** 3

**Summary:**

This paper identifies a dual nature of reasoning in VLMs: while extended reasoning enhances logical capability, it simultaneously weakens visual perception. The authors provide empirical evidence across multiple reasoning benchmarks showing that early-stage reasoning improves accuracy, but performance declines in later stages, with most errors arising from visual perception failures. To address this, the paper first introduces an early-decision mode, where the model is prompted to terminate reasoning mid-trajectory, which mitigates visual forgetting and yields performance gains. Visual Replay and Focus Prompt are also designed to increase attention to visual tokens. Finally, the paper proposes VAPO, a simple yet effective policy-gradient algorithm that inserts visual anchors along the reasoning trajectory to encourage consistent visual perception. Experimental results demonstrate that the proposed VAPO-Thinker-7B improves the model's reliance on visual input and achieves average gains of 2-4% over strong baselines on established benchmarks.

**Strengths:**

- The paper is clearly written and easy to follow.

- The experimental setup is sound and convincingly supports the core claim that prolonged reasoning leads models to neglect visual cues and commit perception-related errors. The quantitative error analysis is comprehensive, showing that perception errors dominate.

- The training framework is effective and well-motivated. I like the design of visual anchors and the perception reward.

**Weaknesses:**

- Since both the insertion of visual anchors and the sampling of reasoning prefixes are random, some anchors may appear in contexts that are unrelated to the visual content described by the corresponding claim, or lack sufficient logical cues for the model to judge the correctness.

- The generation of visual claims relies on GPT-5, which may introduce noise or bias in their quality. This has been pointed out in their limitations.

**Questions:**

- I am interested in the error distribution of the trained VAPO-Thinker. What is the current proportion of perception errors, and do recoverable errors still exist?

- Did the authors verify the reliability of GPT-5–based annotation and claim generation? How was the quality of these generated annotations/claims evaluated?

---

> ### Author Response · Authors · 2025-11-21
> **Response to Reviewer YzHK by Authors (1/2)**
>
> We are grateful for your valuable comments and your acknowledgment of our work. Below are our responses to your concerns.
>
> >**W1: The effect of contextual relevance of visual anchors**
>
> We understand the reviewer’s concern. In fact, during the design of our visual anchor generation and placement strategy, we indeed considered whether the anchors should be explicitly aligned with the semantic context. To examine this, we conducted additional experiments reported in Appendix B.5, where we sampled context-aligned visual anchors by explicitly prompting GPT with both the question and its associated reasoning contexts. We then compared the performance of these context-aligned anchors (QAlignedClaim) with that of our default setting (VisClaim) in Table 6 of the Appendix. For the reviewer’s convenience, we directly present the relevant results here.
>
> |     Method    | MathVerse | LogicVista | Geometry3k |   MMMU   |  MMStar  | HallBench |    Avg   |
> |:-------------:|:---------:|:----------:|:----------:|:--------:|:--------:|:---------:|:--------:|
> | GRPO          |    44.2   |    44.4    |    44.5    |   53.6   |   55.2   |    51.2   |   49.0   |
> | QAlignedClaim |    44.9   |    45.8    |    46.1    |   55.3   |   57.8   |    53.5   |   50.6   |
> | VisClaim      |  **48.9** |  **47.3**  |  **47.7**  | **56.7** | **59.1** |  **55.5** | **52.5** |
>
> As shown in the table above, although these context aligned claims achieve improvements over the GRPO baseline, their performance is instead inferior to that of claims sampled independently of the question. In Section 4.2 of the main paper, we hypothesize that this degradation arises because, when the surrounding reasoning context is strongly correlated with the visual claims, the model may rely primarily on contextual cues to assess claim correctness, thereby reducing its reliance on the visual inputs. This tendency contradicts our motivation of mitigating visual forgetting and ultimately leads to diminished effectiveness. Consequently, our method description in Section 4.2 highlights that one essential requirement for constructing visual claims is their independence from both the question and the reasoning context. This rationale also explains why, by design, we adopt a simple random-insertion strategy rather than context-aligned insertion or other more complex alternatives.
>
> >**W2: The quality of generated visual claims and labels**
>
> Upon obtaining the generated claims, to assess the quality and potential noise of both the visual claims and their corresponding labels, we conducted a straightforward yet reliable human evaluation as an indicator for their effectiveness prior to training. Specifically, we randomly sampled 250 examples from the training set and evaluated each generated claim to determine whether it is *visual-dependent*, i.e., grounded in the visual cues of the image rather than deducible solely through textual reasoning, and whether it is *answerable*, i.e., has a clear and unambiguous ground-truth answer. For the labels, we manually verified the accuracy against the corresponding question and visual input. Below, we report the recorded human evaluation statistics for the claims and labels.
>
> | Human Evaluation | Visual-Dependent (%) | Answerable (%) | Accuracy (%) |
> |:----------------:|:-------------------:|:-------------:|:-----------:|
> |      Claims      |        98.44        |     99.30     |     -     |
> |      Labels      |         -         |      -      |    96.78    |
>
> As shown in the table above, the generated claims exhibit high overall quality according to human evaluation, where the noise in the claims occupy only around 2%, and the labeling error rate is below 4%. In addition, for comprehensiveness, here we also perform model evaluation, in which we sample 2000 examples and feed the generated claims and labels back to GPT and prompt it to act as a judge to assess noise and accuracy as defined above.
>
> | Model Evaluation | Visual-Dependent (%) | Answerable (%) | Accuracy (%) |
> |:----------------:|:-------------------:|:-------------:|:-----------:|
> |      Claims      |        97.29        |     99.67     |     -     |
> |      Labels      |         -         |      -      |    95.85    |
>
>
> The above results indicate that model and human evaluation of claim quality are closely aligned, suggesting that the generated claims and labels are reliable. As highlighted in our limitation analysis, further reducing the noise and error rates has the potential to yield additional performance gains and further enhance the effectiveness of our approach. We have included this experiment in Appendix B.11 of the revised paper (highlighted in blue).

---

> ### Author Response · Authors · 2025-11-21
> **Response to Reviewer YzHK by Authors (2/2)**
>
> >**Q1: The error distribution of our trained models**
>
> In Fig. 12 of the Appendix, we visualized the error distribution of the GRPO model, typically Vision-R1, as well as the corresponding improvements in error rate achieved by our VAPO-Thinker. Here, we report the error rates in numerical formats for both models and, as requested by the reviewer, additionally provide the overall recoverable error ratio across all categories.
>
> | Model        | Perception (%) | Logical (%) | Knowledge (%) | Recoverable (%) |
> |--------------|:--------------:|:-----------:|:-------------:|:---------------:|
> | Vision-R1    |      55.23     |    33.05    |     11.72     |      44.54      |
> | VAPO-Thinker |      36.08     |    25.72    |      7.41     |      18.46      |
>
> As shown in the table, our model substantially reduces the perception error ratio by approximately 20% compared with the GRPO baseline. Moreover, the reduction in the recoverable error ratio is also particularly pronounced, nearly 25%. This indicates that the primary performance gains achieved by our model stems from correcting cases that arise due to perception failures, thereby effectively mitigating visual forgetting.
>
> >**Q2: The evaluation of claim/annotation quality**
>
> Please refer to W2 above for human and model evaluation of claim/annotation quality.

---

> ### Author Response · Authors · 2025-11-28
> **Awaiting Your Valuable Feedback**
>
> Dear reviewer YzHK,
>
> We sincerely appreciate the effort you dedicated to reviewing our paper. Your feedback has greatly enhanced the solidity and clarity of our work. We have provided thorough responses and revisions based on your concerns and suggestions. As the discussion end date approaches, we are eager to know if our revisions have addressed your comments, and we welcome any further questions or suggestions.
>
> Best,
>
> Authors

---

### Author Response · Authors · 2025-11-26
**Summary of Revision**

Dear reviewers and AC,

We are sincerely grateful for the valuable time you dedicated to reviewing our paper and offering insightful feedbacks. We have carefully revised the paper in line with your comments and provided comprehensive responses to all your inquiries. Specifically,

1. Added human and model evaluation of claim/label quality in Section B.11 (reviewer YzHK & j11z)

2. Included efficiency analysis of claim generation in Section B.9 (reviewer 4nqT)

3. Exploring method sensitivity to weaker claim generators other than GPT in Section B.12 (reviewer 4nqT & j11z)

4. Performed training on other base model architectures in Section B.13 (reviewer 4nqT)

5. Conducted evaluations on fine-grained/spatial reasoning tasks in Section B.14 (reviewer 4nqT)

6. Added self-supervised results of our method in Section B.15 (reviewer 4nqT)

7. Provided more baselines for comparison in Section B.20 (reviewer KfZR & j11z)

8. Included discussions about the connections between multiple error categories in Section A.3 (reviewer KfZR)

9. Provided statistics about single-cut recoverable ratio and deduplication checks in Section B.16 and B.17 (reviewer j11z)

10. Added discussion of accuracy gate and anchor placement strategy in Section B.18 and B.19 (reviewer j11z)

12. Added citations of related work in Section 2 (reviewer j11z)

13. Added statistics about long/short thought examples and reasoning length distribution in Section B.21 and B.22 (reviewer j11z)

14. Included evaluation on multi-image settings in Section B.23 (reviewer j11z)

We genuinely hope these revisions have adequately addressed your concerns. As the discussion deadline of **December 2** is approaching, please feel free to reach out if you have any further concerns or suggestions.

Best,

Authors

---

### Author Response · Authors · 2025-11-27
**[Gentle reminder] five days remaining before discussion deadline**

As the discussion deadline (Dec 2, UTC-12) draws near, we kindly invite all reviewers to check out the revised version of the paper and Appendix. We have provided detailed, point-by-point responses to each concern and question. We look forward to hearing whether our revisions have addressed your comments and are open to any further questions or suggestions.

---

### Author Response · Authors · 2025-12-01
**Summary of Rebuttal**

Dear reviewers and AC,

We would like to express our sincere gratitude for the time and effort you dedicated to reviewing our paper. In light of the recent disruptions and the subsequent reassignment of ACs, we would like to offer a brief summary of our rebuttal updates to help facilitate your evaluation of our submission.

**1. Initial Status**

Before the rebuttal period, three of the four reviewers provided highly positive assessments of our submission. Particularly, reviewer YzHK noted that "*the experimental setup is sound and convincingly supports the core claim*" and "*The training framework is effective and well-motivated*", reviewer 4nqT highlighted the "*important and novel problem*", "*strong empirical validation*" and "*elegant method design*", and reviewer j11z emphasized the "*original analysis*", "*methodological novelty*", "*strong empirical results*" and "*efficiency*".

**2. Rebuttal and Discussion**

During the discussion period, we carefully revised the paper in accordance with your comments and provided detailed responses to all inquiries:

* **Claim Generation Process** (reviewer YzHK, 4nqT, j11z): we conducted comprehensive human and model evaluations of claim quality, added an efficiency analysis of the claim generation process, and examined the method sensitivity to weaker claim generators, all of which highlight the effectiveness and robustness of our approach.

* **Additional Results** (reviewer 4nqT, KfZR, j11z): To strengthen the solidity of our approach, as suggested by reviewers, we additionally included results trained on other base models, evaluations on a broader set of tasks, and comparisons with stronger baselines. Results further demonstrate the strong generalization and state-of-the-art performance of our method.

* **Method Extensibility** (reviewer 4nqT, j11z): As recommended by reviewers, we also explored a self-supervised variant of our method that entirely removes the need for external generators, as well as an adaptation to multi-image settings. Both extensions achieved promising results.

* **Supplementary Statistics** (reviewer j11z): As requested, we included extensive supplementary statistics such as single-cut recoverable ratios, data duplication checks, and reasoning length distributions. We also added further ablation studies including anchor placement strategies and accuracy gate design. These results validate the strong motivation, fair evaluation and optimal configurations of our method.

* **Claim justification** (reviewer KfZR): The reviewer raised questions regarding the phrasing and interpretation of several claims. In response, we carefully clarified each issue, refined statements that were overly strong, and included additional quantitative experiments to further substantiate our core contributions.

**3. Conclusion**

For reviewers YzHK, 4nqT, and j11z, we are grateful for the positive recognition they expressed regarding our work, and we believe the new experiments and revisions fully address their concerns, even though they were unable to respond during the discussion period. For reviewer KfZR, after multiple rounds of discussion with additional clarifications and revisions, we are happy that we were able to effectively address their concerns. We appreciate the subsequent score increase **(Scores: 6, 6, 4, 6 → 6, 6, 6, 6)** and the positive feedback "*I agree that the paper offers valuable insights to the community, and I am pleased to see that the authors have adjusted their tone regarding the claim*".

Thank you once again for your contributions throughout the review and discussion process.

Best regards,

Authors

---

### Meta-Review · Area_Chair_QTW3 · 2026-01-08

**Summary:**

The main worries were about rigor of the claims. Reviewers questioned reliance on GPT-5 for claim generation, possible noise or bias in those claims, extra cost and complexity, and whether anchors were placed in a principled way. The biggest concern came from one reviewer who felt the paper overstated “reasoning harms perception” and confused long outputs with genuine reasoning. Others asked for stronger baselines, deduplication checks, and clearer analysis of rewards and efficiency. Overall, the concerns were about framing, reproducibility, and validation rather than whether the method actually works. Most concerns were convincingly addressed after the rebuttal, and thus the AC recommends accept.

**Reviewer Concerns:**

Most concerns were convincingly addressed. The authors added human and model audits of claim quality, showed robustness to weaker open-source generators, provided cost breakdowns, deduplication checks, single-cut recoverable metrics, anchor placement ablations, and new comparisons to strong RL baselines. They also clarified the framing, emphasizing reasoning length and visual forgetting rather than an inherent conflict between reasoning and perception, which resolved the main conceptual objection. What’s still a bit open is separating "reasoning content" from "long context" in a clean way, but reviewers no longer saw this as a blocker.

**Reviewer Scores:**

Reviewer YzHK would almost certainly stay at 6, with higher confidence after seeing the added audits and error breakdowns.
Reviewer 4nqT would also stay at 6, possibly leaning stronger after the new experiments on weaker generators, efficiency, and generalization.
Reviewer j11z would remain at 6, as nearly all requested analyses and checks were added in detail.
Reviewer KfZR explicitly moved from 4 to an intended 6 once framing was softened and baselines were added. Overall, the panel converges to four 6s.

---

### Decision · Program_Chairs · 2026-01-26

Accept (Poster)